# Impact of 16S rRNA Gene Redundancy and Primer Pair Selection on the Quantification and Classification of Oral Microbiota in Next-Generation Sequencing

Alba Regueira-Iglesias,[a] Lara Vázquez-González,[b] Carlos Balsa-Castro,[a] Triana Blanco-Pintos,[a] Nicolás Vila-Blanco,[b] Maria José Carreira,[b] Inmaculada Tomás[a]

[a]Oral Sciences Research Group, Special Needs Unit, Department of Surgery and Medical-Surgical Specialties, School of Medicine and Dentistry, Universidade de Santiago de Compostela, Health Research Institute Foundation of Santiago (FIDIS), Santiago de Compostela, Spain

[b]Centro Singular de Investigación en Tecnoloxías Intelixentes e Departamento de Electrónica e Computación, Universidade de Santiago de Compostela, Health Research Institute Foundation of Santiago (FIDIS), Santiago de Compostela, Spain

**ABSTRACT** This study aimed to evaluate the number of 16S rRNA genes in the complete genomes of the bacterial and archaeal species inhabiting the human mouth and to assess how the use of different primer pairs would affect the detection and classification of redundant amplicons and matching amplicons (MAs) from different taxa. A total of 518 oral-bacterial and 191 oral-archaeal complete genomes were downloaded from the NCBI database, and their complete 16S rRNA genes were extracted. The numbers of genes and variants per genome were calculated. Next, 39 primer pairs were used to search for matches in the genomes and obtain amplicons. For each primer, we calculated the number of gene amplicons, variants, genomes, and species detected and the percentage of coverage at the species level with no MAs (SC-NMA). The results showed that 94.09% of oral bacteria and 52.59% of oral archaea had more than one intragenomic 16S rRNA gene. From 1.29% to 46.70% of bacterial species and from 4.65% to 38.89% of archaea detected by the primers had MAs. The best primers were the following (SC-NMA; region; position for *Escherichia coli* [GenBank version no. J01859 .1]): KP_F048-OP_R030 for bacteria (93.55%; V3 to V7; 342 to 1079), KP_F018-KP_R063 for archaea (89.63%; V3 to V9; undefined to 1506), and OP_F114-OP_R121 for both domains (92.52%; V3 to V9; 340 to 1405). In addition to 16S rRNA gene redundancy, the presence of MAs must be controlled to ensure an accurate interpretation of microbial diversity data. The SC-NMA is a more useful parameter than the conventional coverage percentage for selecting the best primer pairs. The pairs used the most in the oral microbiome literature were not among the best performers.

**IMPORTANCE** Hundreds of publications have studied the oral microbiome through 16S rRNA gene sequencing. However, none have assessed the number of 16S rRNA genes in the genomes of oral microbes, or how the use of primer pairs targeting different regions affects the detection of MAs from different taxa. Here, we found that almost all oral bacteria and more than half of oral archaea have more than one intragenomic 16S rRNA gene. The performance of the primer pairs in not detecting MAs increases as the length of the amplicon augments. As none of those most employed in the oral literature were among the best performers, we selected a series of primers to detect bacteria and/or archaea based on their percentage of species detected without MAs. The intragenomic 16S rRNA gene redundancy and the presence of MAs between distinct taxa need to be considered to ensure an accurate interpretation of microbial diversity data.

**KEYWORDS** 16S rRNA gene, gene variant, matching amplicon, oral microbiota, overestimation factor, primer, redundancy, sequence analysis

Address correspondence to Inmaculada Tomás, inmaculada.tomas@usc.es, or Maria José Carreira, mariajose.carreira@usc.es.

The authors declare no conflict of interest.

The 16S rRNA gene has been widely used to estimate bacterial diversity in different environments (1) ever since its promotion as an "evolutionary clock" some three decades ago (2). This gene, which has an average length of approximately 1,500 base pairs (bps), has several characteristics that have led to its identification as a reliable phylogenetic marker. These are the ubiquitous presence of the 16S ribosomal RNA (rRNA) gene in bacteria and archaea, its relative stability in combining conserved and hypervariable (V) regions, and the existence of complete and easily accessible databases (3).

However, the use of the 16S rRNA gene does not come without limitations, and various investigations have demonstrated the existence of up to 15 gene copies per genome in bacteria (4–8) and up to four in archaea (4, 6, 9). It is well known that this intragenomic gene redundancy affects estimates of microbial abundance that are based on gene counts (4, 7). Overall, there is a tendency for the taxa with a low number of 16S rRNA genes to be underestimated, while those with high numbers are overestimated (7). In addition, the different gene regions do not have the same levels of sequence heterogeneity (6, 10). Meanwhile, V1 to V2 (11), V2 to V3 (12), and V2 and V3 separately (13) have been reported to contain the greatest nucleotide heterogeneity and the greatest discriminatory power between species; other regions, such as V3 to V4 (12) and V4, V5, V7, and V8 (13), have been considered less useful for the same purpose because of their higher degree of sequence conservation. Thus, the primer pair employed in the amplification stage may influence both the detection of redundant amplicons (different 16S rRNA gene amplicons obtained from the same genome) as well as matching amplicons (16S rRNA gene amplicons with a 100% sequence similarity value, MAs) from different taxa.

A recent study has reported the existence of a maximum of four different genes per genome (here, genes/genome) in 32 species isolated from periodontal abscesses (14). However, this limited approach does not reflect the complexity of the oral microbiota, where around 700 species have been identified (15, 16). On the other hand, the identification, at least at the species level, is highly desirable in 16S rRNA sequencing-based studies of the oral microbiome (17). This is because it has been demonstrated how different species from the same genus are associated with different oral conditions (18–20). Our results revealed that *Porphyromonas catoniae* is a core species linked to dental and periodontal health, while *Porphyromonas endodontalis* is associated with dental and periodontal pathology. About the differential abundance data, while *Fusobacterium periodonticum* is present in significantly higher numbers in the dentally healthy, *Fusobacterium nucleatum* subsp. *vicentii* is present in significantly higher numbers in individuals with high degrees of dental pathology (18). However, the taxonomic resolution at the species level could be affected by the presence of MAs.

To the best of our knowledge, there has not yet been an exhaustive *in silico* evaluation of the number of 16S rRNA genes present in the complete genomes of the bacteria and archaea inhabiting the human mouth. Moreover, we have been unable to identify any study in the oral microbiology field that has examined the impact of which primer pair is selected for use to detect and classify redundant amplicons and MAs from different taxa. Consequently, the aims of this investigation were to (i) evaluate the number of 16S rRNA genes in the complete genomes of all the bacterial and archaeal species ever detected in the human oral cavity, using for the first time in the oral microbiome literature the search_16S_py tool (https://github.com/slyalina/search_16S_py), which is based on an algorithm with an estimated sensitivity of >99% for identifying known 16S rRNA genes (21), and (ii) assess how the use of different primer pairs would affect the detection of redundant amplicons and MAs from different taxa.

## RESULTS

**Number of intragenomic 16S rRNA genes in oral-bacterial and oral-archaeal genomes.** Genomes with complete sequencing status in the expanded Human Oral Microbiome Database (eHOMD) (22), along with oral-archaeal complete genomes obtained from data collected in previous research (23), were chosen for inclusion in the present work. A total of 518 oral-bacterial genomes belonging to 186 species, and 191 oral-archaeal

genomes from 135 species, were evaluated. This discrepancy between the number of genomes and species analyzed is because different strains of the same species have different genomes, and some strains have more than one genomic identifier which could belong to chromosomal DNA or plasmids.

Table 1 details the mean number of intragenomic 16S rRNA genes in the bacterial and archaeal phyla through seven taxonomic ranks. The 518 oral-bacterial genomes examined had a mean size of 2,933,660.68 bp and an average number of 4.55 intragenomic 16S rRNA genes, which in turn had a mean size of 1,501.32 bp and an average of 2.60 variants (sequences differing by at least one nucleotide from the reference—the first obtained—sequence). Of the 186 bacterial species, 11 (11/186; 5.91%) had one gene/genome, 159 species (159/186; 85.49%) showed a mean from two to six genes, and 16 species (16/186; 8.60%) had mean values of seven or more genes. The maximum mean number of intragenomic 16S rRNA genes observed was 10.83 in *Bacillus anthracis*, with five strains of this species having a total of 11 genes/genome. Concerning the average number of intragenomic gene variants, 63 bacterial species (63/186; 33.87%) presented one variant/genome, 118 species (118/186; 63.44%), from two to six, and five species (5/186; 2.69%), seven or more.

The 191 oral-archaeal genomes had a mean size of 2,545,441.40 bp and an average of 1.95 intragenomic 16S rRNA genes, which in turn had a mean size of 1,471.25 bp and an average of 1.44 variants. A total of 64 out of the 135 archaeal species (64/135; 47.41%) had a mean of one gene/genome, 67 species (67/135; 49.63%) showed an average from two to three genes, and four species (4/135; 2.96%) had mean values above three (*Methanobacterium formicicum*, *Methanococcus vannielii*, *Methanosphaera stadtmanae*, and *Methanospirillum hungatei*). At the strain level, the maximum total number of genes/genome increased to five in *Methanococcus maripaludis* (unknown strain) and *Sulfolobus acidocaldarius* (unknown strain). Concerning the average number of intragenomic gene variants, 93 species (93/135; 68.89%) had an average number of one variant/genome, and 42 (42/135; 31.11%) had from two to three.

Tables S1 and S2 in the supplemental material contain the sizes of the bacterial and archaeal genomes and genes, the number of genes/genome, and the number of gene variants/genome across eight taxonomic ranks.

**Evaluation of the primer pairs taken from our previous research and those used most in oral microbiome studies.** A total of 33 primer pairs with the best *in silico* coverage values in a previous investigation by our group were selected for evaluation (23). In addition, through a literature review, a list of primer pairs used in oral microbiome studies was compiled in this previous research (23), and the six most frequently repeated (i.e., most commonly used) primers were also selected for evaluation here.

Tables 2 and 3 detail the size and number of 16S rRNA gene amplicons detected by the primer pairs in the oral-bacterial and oral-archaeal genomes. The mean number of 16S rRNA gene amplicons varied from 4.39 to 4.84 for bacteria (mean amplicon variants/genome, 1.09 to 2.69) and 1.58 to 2.43 for archaea (mean amplicon variants/genome, 1.08 to 1.34). All the primer combinations identified the maximum mean numbers of intragenomic genes for the bacterial and archaeal species examined (10.83 and 4.00, respectively). However, although most of the primer pairs were able to detect the highest mean value of the gene variants/genome for the archaeal species (i.e., 3.00), only one primer pair detected this maximum value for bacterial species (i.e., 9.00).

Tables 4 and 5 show the percentages of detected taxa with and without MAs and overabundance estimators obtained by the primer pairs tested on the oral-bacterial and oral-archaeal genomes. Our selected primer pairs detected 16S rRNA gene amplicons in a range of 88.71% (165/186) to 99.46% (185/186) for the bacterial species and 90.37% (122/135) to 99.26% (134/135) for the archaeal species; these percentages were lower for the primer pairs used most in the oral microbiome literature (from 74.19% [138/186] to 95.16% [177/186] - for the bacteria and from 30.37% [41/135] to 63.70% [85/135] for the archaea).

Overall, excluding the most commonly used primer pairs in the literature, unlike the coverage values, the percentage of coverage at the species level with no MAs (SC-NMA)

**TABLE 1** Intragenomic 16S rRNA genes in the bacterial and archaeal phyla through seven taxonomy ranks

| Phylum | Mean no. of intragenomic 16S rRNA genes in each taxonomy level[a] | | | | | | | Mean no. of intragenomic 16S rRNA gene variants[a] | | | | | | | No. of genomes |
|---|---|---|---|---|---|---|---|---|---|---|---|---|---|---|---|
| | Phylum | Class | Order | Family | Genus | Species | Strain | Phylum | Class | Order | Family | Genus | Species | Strain | |
| Actinobacteria | 3.12 | 3.19, 2.00 | 3.41, 1.33 | 4.55, 1.10 | 4.55, 1.00 | 5.00, 1.00 | 5, 1 | 1.54 | 1.56, 1.20 | 2.00, 1.00 | 3.00, 1.00 | 4.00, 1.00 | 4.00, 1.00 | 4, 1 | 91 |
| Bacteroidetes | 3.68 | 4.75, 3.44 | 4.75, 3.44 | 4.75, 2.00 | 4.75, 2.00 | 7.00, 2.00 | 7, 2 | 1.77 | 2.50, 1.61 | 2.50, 1.61 | 2.50, 1.00 | 2.50, 1.00 | 5.00, 1.00 | 5, 1 | 24 |
| C. Saccharibacteria[b] | 1.00 | 1.00 | 1.00 | 1.00 | 1.00 | 1.00 | 1 | 1.00 | 1.00 | 1.00 | 1.00 | 1.00 | 1.00 | 1 | 1 |
| Chlamydiae | 1.00 | 1.00 | 1.00 | 1.00 | 1.00 | 1.00 | 1, 1 | 1.00 | 1.00 | 1.00 | 1.00 | 1.00 | 1.00 | 1, 1 | 5 |
| Chlorobi | 2.00 | 2.00 | 2.00 | 2.00 | 2.00 | 2.00 | 2 | 1.00 | 1.00 | 1.00 | 1.00 | 1.00 | 1.00 | 1 | 1 |
| Chloroflexi | 2.00 | 2.00, 2.00 | 2.00, 2.00 | 2.00, 2.00 | 2.00, 2.00 | 2.00, 2.00 | 2, 2 | 1.50 | 2.00, 1.00 | 2.00, 1.00 | 2.00, 1.00 | 2.00, 1.00 | 2.00, 1.00 | 2, 1 | 2 |
| Firmicutes | 5.43 | 5.52, 3.25 | 6.61, 3.25 | 9.85, 2.00 | 10.18, 2.00 | 10.83, 2.00 | 11, 2 | 3.18 | 3.50, 1.75 | 4.85, 1.75 | 8.00, 1.00 | 9.00, 1.00 | 9.00, 1.00 | 10, 1 | 177 |
| Fusobacteria | 4.35 | 4.35 | 4.35 | 4.40, 4.33 | 4.75, 3.00 | 5.00, 3.00 | 5, 2 | 3.55 | 3.55 | 3.55 | 3.73, 3.00 | 3.73, 3.00 | 5.00, 1.00 | 5, 1 | 21 |
| Ignavibacteriae | 1.00 | 1.00 | 1.00 | 1.00, 1.00 | 1.00, 1.00 | 1.00, 1.00 | 1, 1 | 1.00 | 1.00 | 1.00 | 1.00, 1.00 | 1.00, 1.00 | 1.00, 1.00 | 1, 1 | 2 |
| Proteobacteria | 5.21 | 6.13, 2.17 | 6.98, 2.17 | 7.14, 2.00 | 8.00, 2.00 | 8.00, 2.00 | 8, 2 | 2.87 | 3.55, 1.00 | 4.93, 1.00 | 5.45, 1.00 | 6.17, 1.00 | 8.00, 1.00 | 8, 1 | 170 |
| Spirochaetes | 2.00 | 2.00 | 2.00 | 2.00 | 2.00 | 2.00, 2.00 | 2, 2 | 1.36 | 1.36 | 1.36 | 1.36 | 1.36 | 2.00, 1.22 | 2, 1 | 11 |
| Tenericutes | 1.23 | 1.23 | 1.23 | 1.23 | 1.67, 1.10 | 2.00, 1.00 | 2, 1 | 1.15 | 1.15 | 1.15 | 1.15 | 1.67, 1.00 | 1.67, 1.00 | 2, 1 | 13 |
| C. Thermoplasmatota[b] | 1.00 | 1.00 | 1.00, 1.00 | 1.00, 1.00 | 1.00, 1.00 | 1.00, 1.00 | 1, 1 | 1.00 | 1.00 | 1.00, 1.00 | 1.00, 1.00 | 1.00, 1.00 | 1.00, 1.00 | 1, 1 | 7 |
| Crenarchaeota | 1.10 | 1.10 | 1.25, 1.00 | 1.25, 1.00 | 1.29, 1.00 | 2.00, 1.00 | 5, 1 | 1.00 | 1.00 | 1.00, 1.00 | 1.00, 1.00 | 1.00, 1.00 | 1.00, 1.00 | 1, 1 | 43 |
| Euryarchaeota | 2.29 | 2.67, 1.00 | 2.89, 1.00 | 4.00, 1.00 | 4.00, 1.00 | 4.00, 1.00 | 5, 1 | 1.61 | 1.86, 1.00 | 2.00, 1.00 | 3.00, 1.00 | 3.00, 1.00 | 3.00, 1.00 | 3, 1 | 138 |
| Thaumarchaeota | 1.00 | 1.00 | 1.00 | 1.00 | 1.00 | 1.00, 1.00 | 1, 1 | 1.00 | 1.00 | 1.00 | 1.00 | 1.00 | 1.00, 1.00 | 1, 1 | 3 |

[a]Ranges at the strain level are not mean values, they correspond to the maximum and minimum numbers of intragenomic genes in all strains from a given phylum.

[b]C. Saccharibacteria, *Candidatus* Saccharibacteria; C. Thermoplasmatota, *Candidatus* Thermoplasmatota; No., number.

**TABLE 2** Size and number of 16S rRNA gene amplicons detected by the primer pairs in the oral-bacterial genomes[a,b]

| | | | Superkingdom level | | | Species level | | |
|---|---|---|---|---|---|---|---|---|
| ALC | Primer pair | Gene region | Amplicon length (mean, bp) | g/G (mean) | gv/G (mean) | Amplicon length (range of means, bp) | g/G (range of means) | gv/G (range of means) |
| | Bacterium-specific primer pairs | | | | | | | |
| S | KP_F048-OP_R043 | V3–V4 | 182.99 | 4.62 | 1.25 | 190.00, 162.00 | 10.83, 1.00 | 4.00, 1.00 |
| | OP_F098-OP_R119 | V4–V5 | 288.86 | 4.68 | 1.22 | 290.00, 287.98 | 10.83, 1.00 | 3.00, 1.00 |
| | OP_F066-KP_R040 | V5–V6 | 142.07 | 4.84 | 1.11 | 152.00, 135.00 | 10.83, 1.00 | 2.00, 1.00 |
| | OP_F009-OP_R030 | V5–V7 | 296.13 | 4.60 | 1.38 | 307.00, 283.00 | 10.83, 1.00 | 5.00, 1.00 |
| | KP_F061-KP_R074 | V6–V7 | 206.30 | 4.76 | 1.31 | 212.00, 202.00 | 10.83, 1.00 | 4.00, 1.00 |
| | OP_F101-OP_R030 | V6–V7 | 164.06 | 4.77 | 1.31 | 170.00, 160.00 | 10.83, 1.00 | 3.00, 1.00 |
| M | OP_F053-KP_R020 | V1–V3 | 351.74 | 4.61 | 1.97 | 547.00, 315.00 | 10.83, 1.00 | 8.00, 1.00 |
| | KP_F048-KP_R031 | V3–V5 | 454.84 | 4.61 | 1.42 | 462.00, 433.00 | 10.83, 1.00 | 5.00, 1.00 |
| | KP_F048-OP_R073 | V3–V6 | 546.81 | 4.67 | 1.49 | 554.20, 520.00 | 10.83, 1.00 | 5.00, 1.00 |
| | KP_F051-KP_R041 | V4–V6 | 410.90 | 4.84 | 1.32 | 421.00, 404.00 | 10.83, 1.00 | 3.00, 1.00 |
| | KP_F051-OP_R030 | V4–V7 | 566.17 | 4.62 | 1.55 | 577.00, 552.00 | 10.83, 1.00 | 5.00, 1.00 |
| | OP_F116_KP_R060 | V7–V9 | 308.84 | 4.54 | 1.35 | 1,012.50, 285.00 | 10.83, 1.00 | 4.00, 1.00 |
| L | KP_F048-OP_R030 | V3–V7 | 733.00 | 4.61 | 1.71 | 742.56, 707.00 | 10.83, 1.00 | 5.00, 1.00 |
| | KP_F048-KP_R060 | V3–V9 | 1,059.48 | 4.59 | 1.93 | 1,070.00, 1,016.00 | 10.83, 1.00 | 6.00, 1.00 |
| | KP_F056-KP_R077 | V4–V9 | 846.21 | 4.67 | 1.81 | 1,551.50, 821.00 | 10.83, 1.00 | 6.00, 1.00 |
| | | | | | | | | |
| | Bacterial and archaeal primer pairs | | | | | | | |
| S | OP_F114-KP_R002 | V3–V4 | 188.27 | 4.74 | 1.27 | 194.50, 167.00 | 10.83, 1.00 | 4.00, 1.00 |
| | KP_F020-KP_R032 | V4–V5 | 283.86 | 4.71 | 1.22 | 285.00, 282.98 | 10.83, 1.00 | 3.00, 1.00 |
| | OP_F066-OP_R073 | V5–V6 | 110.11 | 4.65 | 1.09 | 120.00, 101.00 | 10.83, 1.00 | 2.00, 1.00 |
| M | OP_F114-KP_R031 | V3–V5 | 456.84 | 4.60 | 1.42 | 464.00, 435.00 | 10.83, 1.00 | 5.00, 1.00 |
| | OP_F114-OP_R073 | V3–V6 | 548.82 | 4.67 | 1.49 | 556.20, 522.00 | 10.83, 1.00 | 5.00, 1.00 |
| | KP_F020-OP_R073 | V4–V6 | 375.93 | 4.72 | 1.29 | 386.00, 366.00 | 10.83, 1.00 | 3.00, 1.00 |
| L | OP_F114-OP_R121 | V3–V9 | 1060.47 | 4.59 | 1.93 | 1,071.00, 1017.00 | 10.83, 1.00 | 6.00, 1.00 |
| | KP_F020-OP_R121 | V4–V9 | 889.30 | 4.70 | 1.82 | 1,594.50, 864.00 | 10.83, 1.00 | 6.00, 1.00 |
| | OP_F066-OP_R121 | V5–V9 | 623.05 | 4.55 | 1.65 | 1,328.50, 598.00 | 10.83, 1.00 | 6.00, 1.00 |
| | | | | | | | | |
| | Most-used primer pairs | | | | | | | |
| S | KP_F078-OP_R010[B+A] | V4–V5 | 291.86 | 4.71 | 1.22 | 293.00, 290.98 | 10.83, 1.00 | 3.00, 1.00 |
| M | KP_F031-KP_R021[B] | V1–V4 | 525.59 | 4.39 | 2.03 | 700.00, 467.00 | 10.83, 1.00 | 8.00, 1.00 |
| | KP_F047-KP_R035[B] | V3–V5 | 460.25 | 4.61 | 1.42 | 467.00, 438.00 | 10.83, 1.00 | 5.00, 1.00 |
| | OP_F009-OP_R029[B] | V5–V8 | 409.87 | 4.76 | 1.51 | 417.00, 395.00 | 10.83, 1.00 | 5.00, 1.00 |
| L | KP_F034-KP_R065[B] | V1–V9 | 1505.85 | 4.81 | 2.69 | 1,677.00, 1429.00 | 10.83, 1.00 | 9.00, 1.00 |

[a]The amplicon length category and gene regions were determined in a previous investigation according to the mean size of the amplicons generated by a given primer and to the mode first position of the forward primer and the mode last position of the reverse primer, respectively (23). The most commonly used primer pairs in the literature were detected in the before-mentioned study (23). In the column "gene region", the different regions covered by each primer pair are defined and delimited by a dash.
[b]A, archaea; ALC, amplicon length category; B, bacteria; bp(s), base pair(s); F, forward; g/G, number of 16S rRNA gene amplicons per genome; gv/G, number of 16S rRNA gene variant amplicons per genome; KP, Klindworth primer; L, long mean amplicon length category ($>$600 bp); M, medium mean amplicon length category (301 to 600 bp); OP, oral primer; R, reverse; S, short mean amplicon length category, (100 to 300 bp); V, hypervariable region.

increased as the mean length of the amplicons obtained by the primer pair increased. If we contrast the percentages of species detected with their respective SC-NMA, all the primer pairs with short mean amplicon lengths (S; 100 to 300 bp) analyzed showed the largest differences between both parameters (average difference, 21.34% for bacteria and 23.70% for archaea), followed by those of medium mean amplicon lengths (M; 301 to 600 bp; 7.30% and 13.75%, respectively). The primer pairs with long mean amplicon lengths (L; $>$600 bp) presented the smallest differences between the coverage and SC-NMA values (4.30% and 5.82%, respectively).

According to the SC-NMA values, the best three bacteria-specific primer pairs were KP_F048-OP_R030 and KP_F048-KP_R060 (L; regions V3 to V7 and V3 to V9, respectively; SC-NMA, 93.55%; six MAs; overestimation factor caused by the presence of MAs [OF-MA], 1.06 for both) and OP_F053-KP_R020 (M, V1 to V3, 93.01%, six, 1.06). In contrast, the worst primer pair was OP_F066-KP_R040 (S, V5 to V6, 47.31%, 77, 2.78). The most commonly used bacterium-specific primer pairs in the literature were not among those with the best SC-NMA values in their respective amplicon length categories. Specifically, KP_F031-KP_R021 (M, V1 to V4, 73.12%, two, 1.02) and OP_F009-OP_R029 (M, V5 to V8, 75.27%, 24, 1.24) had the worst SC-NMA among the primers in the

**TABLE 3** Size and number of 16S rRNA gene amplicons detected by the primer pairs in the oral-archaeal genomes[a,b]

| ALC | Primer pair | Gene region | Superkingdom level | | | Species level | | |
|---|---|---|---|---|---|---|---|---|
| | | | Amplicon length (mean, bp) | g/G (mean) | gv/G (mean) | Amplicon length (range of means, bp) | g/G (range of means) | gv/G (range of means) |
| | Archaeal-specific primer pairs | | | | | | | |
| S | KP_F018-KP_R002 | V4 | 154.68 | 1.96 | 1.09 | 860.00, 138.00 | 4.00, 1.00 | 3.00, 1.00 |
| | OP_F066-KP_R013 | V5–V6 | 274.32 | 1.99 | 1.11 | 277.00, 274.00 | 4.00, 1.00 | 2.00, 1.00 |
| M | KP_F018-KP_R032 | V3–V5 | 429.16 | 1.95 | 1.18 | 1914.00, 407.00 | 4.00, 1.00 | 3.00, 1.00 |
| | KP_F018-OP_R073 | V3–V5 | 526.11 | 1.98 | 1.22 | 2010.00, 503.00 | 4.00, 1.00 | 3.00, 1.00 |
| | KP_F020-KP_R013 | V3–V6 | 545.97 | 1.99 | 1.21 | 1325.00, 539.00 | 4.00, 1.00 | 3.00, 1.00 |
| | KP_F022-KP_R063 | V5–V9 | 586.51 | 2.00 | 1.22 | 670.00, 530.00 | 4.00, 1.00 | 3.00, 1.00 |
| L | OP_F114-KP_R013 | V3–V6 | 693.70 | 1.99 | 1.26 | 2178.00, 671.00 | 4.00, 1.00 | 3.00, 1.00 |
| | KP_F018-KP_R063 | V3–V9 | 1,131.87 | 1.96 | 1.34 | 1828.00, 1056.00 | 4.00, 1.00 | 3.00, 1.00 |
| | OP_F066-OP_R016 | V5–V9 | 623.27 | 2.01 | 1.22 | 626.33, 620.00 | 4.00, 1.00 | 3.00, 1.00 |
| | Bacterial and archaeal primer pairs | | | | | | | |
| S | OP_F114-KP_R002 | V3–V4 | 162.29 | 1.95 | 1.10 | 868.00, 146.00 | 4.00, 1.00 | 3.00, 1.00 |
| | KP_F020-KP_R032 | V4–V5 | 289.43 | 1.95 | 1.14 | 1,069.00, 283.00 | 4.00, 1.00 | 3.00, 1.00 |
| | OP_F066-OP_R073 | V5–V6 | 114.03 | 1.98 | 1.08 | 1,15.00, 114.00 | 4.00, 1.00 | 2.00, 1.00 |
| M | OP_F114-KP_R031 | V3–V5 | 436.72 | 1.95 | 1.19 | 1,922.00, 415.00 | 4.00, 1.00 | 3.00, 1.00 |
| | OP_F114-OP_R073 | V3–V6 | 533.62 | 1.98 | 1.23 | 2,018.00, 511.00 | 4.00, 1.00 | 3.00, 1.00 |
| | KP_F020-OP_R073 | V4–V6 | 385.76 | 1.99 | 1.18 | 1,165.00, 379.00 | 4.00, 1.00 | 3.00, 1.00 |
| L | OP_F114-OP_R121 | V3–V9 | 1042.81 | 1.98 | 1.33 | 1,741.00, 1023.00 | 4.00, 1.00 | 3.00, 1.00 |
| | KP_F020-OP_R121 | V4–V9 | 907.41 | 1.98 | 1.29 | 2,279.00, 891.00 | 4.00, 1.00 | 3.00, 1.00 |
| | OP_F066-OP_R121 | V5–V9 | 635.78 | 1.98 | 1.22 | 1,323.00, 625.00 | 4.00, 1.00 | 3.00, 1.00 |
| | Most-used primer pairs | | | | | | | |
| S | KP_F078-OP_R010[B+A] | V4–V5 | 292.79 | 2.43 | 1.21 | 294.00, 291.50 | 4.00, 1.00 | 3.00, 1.00 |
| L | KP_F014-KP_R011[A] | V3–V6 | 606.18 | 1.58 | 1.14 | 603.00, 608.00 | 4.00, 1.00 | 3.00, 1.00 |

[a]The amplicon length category and gene regions were determined in a previous investigation according to the mean size of the amplicons generated by a given primer and to the mode first position of the forward primer and the mode last position of the reverse primer, respectively (23). The most commonly used primer pairs in the literature were detected in the before-mentioned study (23). In the column "gene region", the different regions covered by each primer pair are defined and delimited by a dash.
[b]A, archaea; ALC, amplicon length category; B, bacteria; bp(s), base pair(s); F, forward; g/G, number of 16S rRNA gene amplicons per genome; gv/G, number of 16S rRNA gene variant amplicons per genome; KP, Klindworth primer; L, long mean amplicon length category (>600 bp); M, medium mean amplicon length category (301 to 600 bp); OP, oral primer; R, reverse; S, short mean amplicon length category (100 to 300 bp); V, hypervariable region.

medium amplicon length category, and KP_F034-KP_R065 (L, V1 to V9, 82.26%, two, 1.02) had the worst value among those in the long category.

Considering the three categories of amplicon lengths, the SC-NMA values for the archaeon-specific primer pairs were 89.63% for KP_F018-KP_R063 (L; V3 to V9; six MAs; OF-MA, 1.11), 85.93% for KP_F022-KP_R063 (M, V5 to V9, eight, 1.14), and 69.63% for the OP_F066-KP_R013 (S, 35, 1.99). Interestingly, the primer pair with a long mean amplicon length, KP_F014-KP_R011 (V3 to V6), which is the one used most in the literature to detect oral archaea, was only able to identify 30.37% of the species tested in this study, resulting in the lowest SC-NMA value (26.67%; five MAs; OF-MA, 1.14).

In relation to the bacterial and archaeal primer pairs, the overall SC-NMA values were 92.52% for OP_F114-OP_R121 (L; V3 to V9; 12 MAs; OF-MA, 1.08), 88.79% for OP_F114-KP_R031 (M, V3 to V5, 29, 1.26), and 54.21% for OP_F066-OP_R073 (S, V5 to V6, 134, 3.45). In terms of overall SC-NMA, the second worst was KP_F078-OP_R010 (S; V4 to V5; 66.67%; 48 MAs; OF-MA, 1.68), mainly due to its low capacity to detect archaea (63.70%), which directly affected the SC-NMA value for archaea (48.89%) (Table 5). However, this primer pair is the most widely used in the literature to detect bacteria and archaea.

Tables S3 to S6 contain more detailed information on the results of MA- and MA-free species coverage and the overabundance parameters (overestimation factor [OF] and OF-MA values) obtained by the primer pairs tested against the oral-bacterial and oral-archaeal genomes. Tables S3 and S6 also include the results obtained from the bacterial and archaeal primer pairs for both domains.

Depending on the primer pair tested, from 1.29% (2/155 detected species) to 46.70% (85/182) of the bacterial species and from 4.65% (6/129) to 38.89% (49/126) of the archaeal species had MAs (Tables 4 and 5). Fig. 1 shows the bacterial and archaeal

**TABLE 4** Detected taxa with and without matching amplicons and overabundance estimators obtained by the primer pairs tested on the oral-bacterial genomes[a,b]

| ALC | Primer pair | No. of detected genomes (%) | No. of detected species (%) | No. of detected species with MAs (%) | No. of SC-NMA (%) | No. of OF | No. of OF-MA |
|---|---|---|---|---|---|---|---|
| | Bacterium-specific primer pairs | | | | | | |
| S | KP_F048-OP_R043 | 508 (98.07) | 180 (96.77) | 22 (12.22) | 158 (84.95) | 5.82 | 1.30 |
| | OP_F098-OP_R119 | 493 (95.17) | 177 (95.16) | 28 (15.82) | 149 (80.11) | 8.28 | 1.73 |
| | OP_F066-KP_R040 | 455 (87.84) | 165 (88.71) | 77 (46.67) | 88 (47.31) | 13.16 | 2.78 |
| | OP_F009-OP_R030 | 504 (97.30) | 181 (97.31) | 29 (16.02) | 152 (81.72) | 6.01 | 1.32 |
| | KP_F061-KP_R074 | 468 (90.35) | 169 (90.86) | 39 (23.08) | 130 (69.89) | 7.48 | 1.61 |
| | OP_F101-OP_R030 | 460 (88.80) | 167 (89.78) | 39 (23.35) | 128 (68.82) | 7.44 | 1.61 |
| M | OP_R053-KP_R020 | 506 (97.68) | 179 (96.24) | 6 (3.35) | 173 (93.01) | 4.80 | 1.06 |
| | KP_F048-KP_R031 | 507 (97.88) | 180 (96.77) | 9 (5.00) | 171 (91.94) | 5.04 | 1.12 |
| | KP_F048-OP_R073 | 498 (96.14) | 178 (95.70) | 6 (3.37) | 172 (92.47) | 4.72 | 1.06 |
| | KP_F051-KP_R041 | 456 (88.03) | 166 (89.25) | 20 (12.05) | 146 (78.50) | 7.45 | 1.58 |
| | KP_F051-OP_R030 | 508 (98.07) | 184 (98.92) | 19 (10.33) | 165 (88.71) | 5.53 | 1.22 |
| | OP_F116_KP_R060 | 516 (99.61) | 185 (99.46) | 31 (16.76) | 154 (82.80) | 6.13 | 1.37 |
| L | KP_F048-OP_R030 | 507 (97.88) | 180 (96.77) | 6 (3.33) | 174 (93.55) | 4.72 | 1.06 |
| | KP_F048-KP_R060 | 507 (97.88) | 180 (96.77) | 6 (3.33) | 174 (93.55) | 4.72 | 1.06 |
| | KP_F056-KP_R077 | 495 (95.56) | 180 (96.77) | 10 (5.56) | 170 (91.40) | 4.89 | 1.10 |
| | Bacterial and archaeal primer pairs | | | | | | |
| S | OP_F114-KP_R002 | 485 (93.63) | 172 (92.47) | 22 (12.79) | 150 (80.65) | 5.82 | 1.30 |
| | KP_F020-KP_R032 | 488 (94.21) | 176 (94.62) | 28 (15.91) | 148 (79.57) | 8.28 | 1.73 |
| | OP_F066-OP_R073 | 502 (96.91) | 182 (97.85) | 85 (46.70) | 97 (52.15) | 15.90 | 3.31 |
| M | OP_F114-KP_R031 | 507 (97.88) | 180 (96.77) | 9 (5.00) | 171 (91.94) | 5.04 | 1.12 |
| | OP_F114-OP_R073 | 498 (96.14) | 178 (95.70) | 6 (3.77) | 172 (92.47) | 4.72 | 1.06 |
| | KP_F020-OP_R073 | 488 (94.21) | 176 (94.62) | 22 (12.50) | 154 (82.80) | 7.52 | 1.61 |
| L | OP_F114-OP_R121 | 507 (97.88) | 180 (96.77) | 6 (3.33) | 174 (93.55) | 4.72 | 1.06 |
| | KP_F020-OP_R121 | 489 (94.40) | 177 (95.16) | 10 (5.65) | 167 (89.79) | 4.89 | 1.10 |
| | OP_F066-OP_R121 | 516 (99.61) | 185 (99.46) | 16 (8.65) | 169 (90.86) | 5.20 | 1.16 |
| | Most-used primer pairs | | | | | | |
| S | KP_F078-OP_R010[B+A] | 488 (94.21) | 176 (94.62) | 28 (15.91) | 148 (79.57) | 8.28 | 1.73 |
| M | KP_F031-KP_R021[B] | 347 (66.99) | 138 (74.19) | 2 (1.45) | 136 (73.12) | 4.50 | 1.02 |
| | KP_F047-KP_R035[B] | 500 (96.53) | 177 (95.16) | 9 (5.09) | 168 (90.32) | 5.04 | 1.12 |
| | OP_F009-OP_R029[B] | 469 (90.54) | 164 (88.17) | 24 (14.63) | 140 (75.27) | 5.62 | 1.24 |
| L | KP_F034-KP_R065[B] | 440 (84.94) | 155 (83.33) | 2 (1.29) | 153 (82.26) | 4.50 | 1.02 |

[a]Percentage calculations were performed taking into account the total number of genomes (518) and bacterial species (186), except for the variable "detected species with MAs," in which the number of species detected by a given primer was considered.

[b]A, archaea; ALC, amplicon length category; B, bacteria; F, forward; KP, Klindworth primer; L, long mean amplicon length category (>600 bp); M, medium mean amplicon length category (301 to 600 bp); MAs, matching amplicons; No., number; OF, overestimation factor; OF-MA, overestimation factor associated with matching amplicons; OP, oral primer; R, reverse; S, short mean amplicon length category (100 to 300 bp); SC-NMA, species coverage with no matching amplicons.

species with MAs obtained with at least 10 primer pairs. In the bacteria, these species belonged to the following genera: *Actinomyces*, *Cronobacter*, *Fusobacterium*, *Klebsiella*, *Lacticaseibacillus*, *Lactobacillus*, *Staphylococcus*, and *Streptococcus*. In the archaea, these genera were *Haloarcula*, *Halomicrobium*, *Methanosarcina*, *Nitrososphaera*, *Pyrococcus*, and *Thermococcus*. Tables S7 to S9 define in detail which species from the same or different genera shared MAs, depending on the primer pair evaluated.

## DISCUSSION

**Number of intragenomic 16S rRNA genes in oral-bacterial and oral-archaeal genomes.** The intragenomic redundancy of the 16S rRNA gene has been evaluated previously using genomes from diverse sources such as GenBank (5, 7, 24, 25), the National Center for Biotechnology Information (NCBI) microbial genome database (4, 6, 8, 26), and the rRNA Operon Copy Number Database (rrnDB) (9, 27, 28). These *in silico* investigations extracted the gene sequences from the complete genomes through tools such as Kodon 2.0 (24; https://www.bionumerics.com/download/software/kodon -version-204) and RNAmmer (6, 29) or by using a primer pair targeting the regions V4 to V6 (7). However, none of these studies focused on the genomes of microorganisms living in a specific environment. As gene redundancy has been proven to affect abundance estimates based on gene counts (4, 7), variations in the number of genes/genome

**TABLE 5** Detected taxa with and without matching amplicons and overabundance estimators obtained by the primer pairs tested on the oral-archaeal genomes[a,b]

| ALC | Primer pair | No. of detected genomes (%) | No. of detected species (%) | No. of detected species with MAs (%) | No. of SC-NMA (%) | No. of OF | No. of OF-MA |
|---|---|---|---|---|---|---|---|
| | Archaeal-specific primer pairs | | | | | | |
| S | KP_F018-KP_R002 | 185 (96.86) | 129 (95.56) | 29 (22.48) | 100 (74.07) | 3.30 | 1.76 |
| | OP_F066-KP_R013 | 184 (96.34) | 129 (95.56) | 35 (27.13) | 94 (69.63) | 4.02 | 1.99 |
| M | KP_F018-KP_R032 | 186 (97.38) | 130 (96.30) | 20 (15.39) | 110 (81.48) | 2.68 | 1.49 |
| | KP_F018-OP_R073 | 177 (92.67) | 122 (90.37) | 18 (14.75) | 104 (77.04) | 2.65 | 1.46 |
| | KP_F020-KP_R013 | 183 (95.81) | 128 (94.81) | 20 (15.63) | 108 (80.00) | 2.61 | 1.35 |
| | KP_F022-KP_R063 | 180 (94.24) | 124 (91.85) | 8 (6.45) | 116 (85.93) | 2.26 | 1.14 |
| L | OP_F114-KP_R013 | 184 (96.34) | 129 (95.56) | 16 (12.40) | 113 (83.70) | 2.47 | 1.28 |
| | KP_F018-KP_R063 | 183 (95.81) | 127 (94.07) | 6 (4.72) | 121 (89.63) | 2.16 | 1.11 |
| | OP_F066-OP_R016 | 180 (94.24) | 124 (91.85) | 8 (6.45) | 116 (85.93) | 2.26 | 1.14 |
| | Bacterial and archaeal primer pairs | | | | | | |
| S | OP_F114-KP_R002 | 190 (99.48) | 134 (99.26) | 29 (21.64) | 105 (77.78) | 3.30 | 1.76 |
| | KP_F020-KP_R032 | 190 (99.48) | 134 (99.26) | 30 (22.39) | 104 (77.04) | 3.88 | 1.92 |
| | OP_F066-OP_R073 | 181 (94.76) | 126 (93.33) | 49 (38.89) | 77 (57.04) | 8.37 | 3.71 |
| M | OP_F114-KP_R031 | 190 (99.48) | 134 (99.26) | 20 (14.93) | 114 (84.44) | 2.68 | 1.49 |
| | OP_F114-OP_R073 | 181 (94.76) | 126 (93.33) | 18 (14.29) | 108 (80.00) | 2.65 | 1.46 |
| | KP_F020-OP_R073 | 180 (94.24) | 125 (92.59) | 26 (20.80) | 99 (73.33) | 3.74 | 1.85 |
| L | OP_F114-OP_R121 | 185 (96.86) | 129 (95.56) | 6 (4.65) | 123 (91.11) | 2.16 | 1.11 |
| | KP_F020-OP_R121 | 185 (96.86) | 129 (95.56) | 6 (4.65) | 123 (91.11) | 2.16 | 1.11 |
| | OP_F066-OP_R121 | 186 (97.38) | 130 (96.30) | 8 (6.15) | 122 (90.37) | 2.26 | 1.14 |
| | Most-used primer pairs | | | | | | |
| S | KP_F078-OP_R010[B+A] | 123 (66.40) | 86 (63.70) | 20 (23.26) | 66 (48.89) | 3.56 | 1.60 |
| L | KP_F014-KP_R011[A] | 44 (23.04) | 41 (30.37) | 5 (12.20) | 36 (26.67) | 2.00 | 1.14 |

[a]Percentage calculations were performed taking into account the total number of genomes (191) and archaeal species (135), except for the variable "detected species with MAs," in which the number of species detected by a given primer was considered.

[b]A, archaea; ALC, amplicon length category; B, bacteria; F, forward; KP, Klindworth primer; L, long mean amplicon length category (>600 bp); M, medium mean amplicon length category (301 to 600 bp); MAs, matching amplicons; No., number; OF, overestimation factor; OF-MA, overestimation factor associated with matching amplicons; OP, oral primer; R, reverse; S, short mean amplicon length category (100 to 300 bp); SC-NMA, species coverage with no matching amplicons.

of the microbes inhabiting the ecosystem of interest must be examined to ensure proper descriptions of the microbial community. To the best of our knowledge, this study is the first to investigate the number of intragenomic 16S rRNA genes in the microbiota of the oral environment.

Through chromatograms derived from direct sequencing or cloning, recent research identified a maximum of four different 16S rRNA genes/genome in 138 clinical isolates taken from periodontal abscesses (14). However, the low number of species evaluated ($n = 32$) and the focus on a specific niche and health condition within the mouth limit the applicability of the findings to the oral microbiota more generally. In contrast, the present study evaluated all of the complete bacterial genomes described in an oral-specific database (22) and a series of genomes taken from archaeal species previously identified in the human mouth (23); all these bacterial and archaeal genomes were downloaded from the NCBI website (26). Moreover, for the first time in the oral microbiome literature, we extracted the 16S rRNA genes using search_16S_py (https://github.com/slyalina/search_16S_py), a special and easily accessible tool based on Edgar's algorithm, which has an estimated sensitivity of >99% for identifying known 16S rRNA genes (21). In our opinion, this algorithm represents a significant improvement in the detection of the 16S rRNA genes over previously used methods (7, 24) since it constitutes a specialized tool for this purpose.

Our study identified that 94.09% of the oral-bacterial species had more than one 16S rRNA gene/genome and 8.60% had seven or more, which are values similar to those previously reported in non-oral-specific investigations (95.53% and ~9.50%, respectively) (5, 7). Conversely, other authors found greater percentages of bacteria with one intragenomic gene (15.00% versus 5.91% in the present study) (4, 7) or with seven or more (17.80% versus 8.60% in the present study) (4). Also, we detected that 47.41% of the oral-archaeal species had one gene/genome, which is considerably lower than reported before in nonoral studies (65.20% and 57.00%) (4, 9). Consequently, we

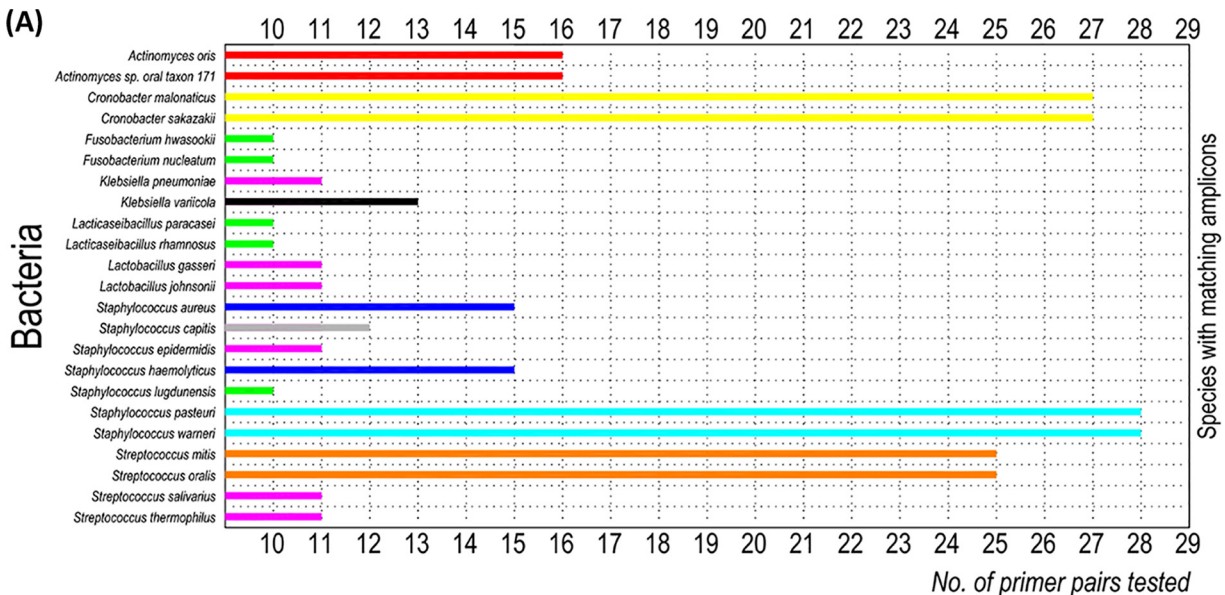

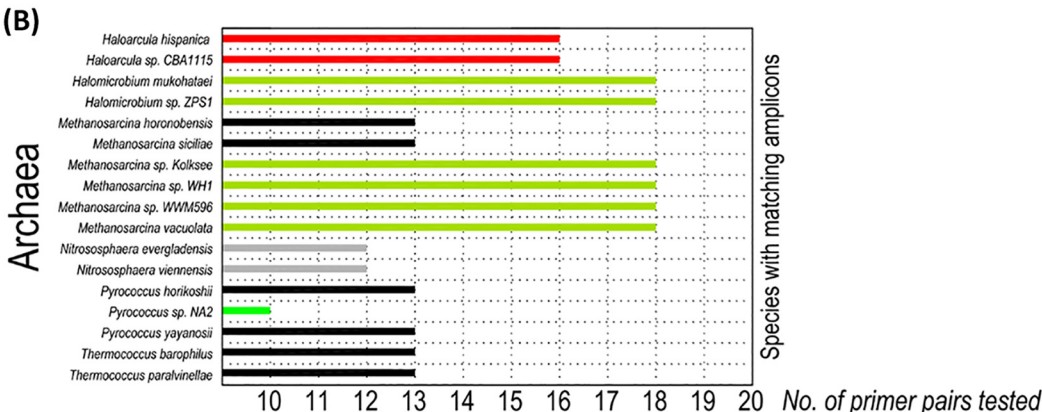

**FIG 1** (A and B) Taxa with matching amplicons using at least 10 primer pairs: bacterial species (A) and archaeal species (B). To detect bacteria and archaea, 29 and 20 primer pairs, respectively, were tested, including domain-specific primers, those (primers) suitable for bacteria and archaea, and those (primers) widely used in the literature.

found that several species traditionally associated with different oral-health conditions had more than one intragenomic 16S rRNA gene, meaning they may have been overcounted in previous sequencing-based investigations. Included in these species were bacteria that are widely known to be associated with periodontitis, such as *Aggregatibacter actinomyce-temcomitans* (mean genes/genome, 5.75), *F. nucleatum* (4.25), *Filifactor alocis* (4.00), *Porphyromonas gingivalis* (4.00), *Tannerella forsythia* (2.00), and *Treponema denticola* (2.00) (30); the caries-associated bacteria *Streptococcus mutans* (5.00) (31) and *Rothia dentocariosa* (3.00) (32); and the commensal bacteria *Streptococcus mitis* (4.00) (33) and *Streptococcus oralis* (4.00) (34). Some archaeal species that can be found in healthy subjects or those with periodontitis (35) also had more than one gene/genome. These included *M. stadtmanae* (4.00), *Methanosarcina mazei* (3.00), *M. maripaludis* (2.88), *Methanobrevibacter smithii* (2.00), and *S. acidocaldarius* (2.00).

Still, it has to be said that the above-referenced sequencing-based research (30–35) did not aim to account for or correct for the variation in the 16S rRNA gene copy numbers between taxa, nor did they point out that this was a study limitation. Indeed, as far as we know, no investigation of the oral microbiome has used the methods available to correct for the variation in the numbers of 16S rRNA genes, such as CopyRighter (36), PAPRICA (37), or PICRUSt (38). Given the results obtained in the present study, we encourage researchers to incorporate these tools in their clinical investigations as one of

the necessary steps to obtain a picture of the oral microbiome composition as close to reality as possible.

**Evaluation of the primer pairs taken from our previous research and those used most in oral microbiome studies.** There is a lack of literature on how the 16S rRNA gene primer pair influences the detection of redundant amplicons and MAs from different taxa. Recognizing the importance of conducting 16S rRNA gene-based research using habitat-specific databases (17), the present study is the first to evaluate the above-mentioned topic focusing on the oral microbiota.

Through this analysis, we discovered that all the primer combinations amplified the maximum mean number of genes/genome in both the bacterial and archaeal species (10.83 and 4.00, respectively). However, the great majority of them could not detect the maximum mean number of variants/genome in bacteria (i.e., 9.00), which was not the case for the archaea.

The presence of amplicons with matching 16S rRNA gene sequences in different species is a challenge for researchers, as they could be inappropriately misclassified, thus artificially increasing the number of counts in operational taxonomic units (OTUs) or amplicon sequence variants (ASVs) (7, 39), depending on the bioinformatics pipeline used. As amplicons derived from distinct regions have different degrees of heterogeneity (6, 10), with V1 to V2 (11), V2 to V3 (12), V2, and V3 (13) having more nucleotide variability than V3 to V4 (12), V4, V5, V7, and V8 (13), the primer pair employed may affect both estimates of diversity and taxonomic identifications. Despite the lack of literature on the subject, it is important from a clinical applicability point of view to conduct studies that take into account the quality of the primer pairs, in our case those specific to the oral microbiota.

As described in a previous investigation conducted by our research team (23), the primer pairs that identified >90% of the species in a data set were evenly distributed across the different amplicon length categories. However, these findings do not reflect the influence of MAs. To ensure that this factor was taken into account in the present study, we considered, for the first time in this type of research, the values of the percentage of coverage at the species level with no matching amplicons (SC-NMA), the overestimation factor (OF), which combines the copy number of the 16S rRNA gene amplicons and the number of MAs, and the OF caused by the presence of MAs (OF-MA). The lack of studies employing these parameters makes it impossible to conduct a relevant comparative analysis.

The estimation tools that we newly describe have allowed us to demonstrate in general terms that the primer pairs that obtain amplicons with long mean lengths (>600 bp), followed by those with medium mean lengths (301 to 600 bp), showed the greatest ability to detect bacterial and archaeal species with no MAs in contrast to primer pairs that obtain amplicons with short mean lengths (100 to 300 bp), being the mean differences between coverage and SC-NMA percentages of 22.52%, 10.52%, and 5.06%, respectively. These discrepancies between the two coverage parameters were confirmed by considering the coverage results of a previous study in which we analyzed a larger number of oral taxa based on 16S rRNA gene sequences instead of complete genomes (23). Assessing the impact that MAs could have on species abundance, primer pairs with long mean amplicon lengths had OF-MA values as low as 1.08 (e.g., with OP_F114-OP_R121, V3 to V9), meaning that the small number of MAs detected did not influence abundance. In contrast, primer pairs with short mean amplicon lengths had a maximum value of 3.45 (e.g., with OP_F066-OP_R073, V5 to V6), meaning that abundance was tripled by the presence of MAs.

The above findings reveal that the SC-NMA parameter is more useful than the conventional coverage percentage in selecting the best primer pairs because this last value does not discriminate the presence of MAs for different taxa. If there are several primer pairs with similar SC-NMA values, the OF-MA values would be the appropriate parameter to use to choose between them.

Specifically, the best primer pairs presented mean amplicon lengths of >600 bp and were KP_F048-OP_R030 (for bacteria [B]; V3 to V7; SC-NMA, 93.55%; OF-MA, 1.06),

KP_F018-KP_R063 (for archaea [A], V3 to V9, 89.63%, 1.11); and OP_F114-OP_R121 (for bacteria and archaea jointly [B+A], V3 to V9, 92.52%, 1.08). In consequence, we were thus able to demonstrate that sequencing longer fragments enables the identification of lower taxonomy levels (40), reducing the probability of overestimation and classification bias related to MAs. This finding connects with the idea that sequencing the full-length 16S rRNA gene has been regarded as the better solution to the limitations of taxonomic classification (11, 40) since it would at least avoid taxonomic classification biases related to MAs in oral bacteria and archaea. However, this assumption is only applicable when using sequence classification methods based on single-nucleotide resolution (100% sequence similarity) that generate ASVs. Considering that the taxonomic threshold of sequence similarity to define the species level has been set from 97% to 98.7% (11, 41), we recently performed an *in silico* study on oral bacteria and archaea and found that the primer pairs with long mean amplicon lengths exhibit species coverage with no amplicon sequence similarity ≥97% (SC-NASI97) values much lower than those observed here for SC-NMA (up to 51.08% for bacteria, 49.63% for archaea, and 48.29% for both domains) (42). Therefore, full-length gene sequencing will not avoid taxonomic classification biases if we use sequence clustering methods based on 97% sequence similarity thresholds, as is commonly done for OTU construction. On the other hand, we must also take into account that the technologies which currently allow sequencing of the full-length 16S rRNA gene, such as Pacific Biosciences (PacBio) and Oxford Nanopore Technologies (ONT), still have high error rates (10% to 15%) (43).

None of the pairs used most in the oral microbiome sequencing-based studies reported in the literature were able to detect the highest possible numbers of total genomes and species. We might assume *a priori* that the lower the number of taxa detected, the fewer the opportunities to misclassify them, but the best SC-NMA estimates were also not obtained with these primer pairs. The pair employed most in the literature is KP_F078-OP_R010 (B+A, V4 to V5) (23), as described by Caporaso et al. (44). This showed an SC-NMA score of 66.67% and an OF-MA of 1.68 and was the second-worst primer pair at detecting both the bacterial and archaeal superkingdoms. It was even outperformed by other primers from the same region, such as OP_F098-OP_R119 (B; V4 to V5; SC-NMA, 80.11%; OF-MA, 1.73) if only bacteria were to be detected and KP_F020-KP_R032 (B+A, V4 to V5, 78.51%, 1.79) if bacteria and archaea were to be detected. The next most-used primer pair (23) is the one recommended by Illumina (45): KP_F047-KP_R035 (B; V3 to V5; SC-NMA, 90.32%; OF-MA, 1.12). Although it produced an SC-NMA value of >90%, this was poorer than another primer targeting the same gene region: KP_F048-KP_R031 (B, V3 to V5, 91.94%, 1.12). The other primer pairs used most in the literature, albeit to a lesser extent (23), are KP_F014-KP_R011 (A; V3 to V6; SC-NMA, 26.67%; OF-MA, 1.14), KP_F034-KP_R065 (B, V1 to V9, 82.26%, 1.02), KP_F031-KP_R021 (B, V1 to V4, 73.12%, 1.02), and OP_F009-OP_R029 (B, V5 to V8, 75.27%, 1.24). Of them, the first has produced the lowest SC-NMA value reported here, which is considerably lower than the 80.00% achieved by the primer pair from the same region, KP_F020-KP_R013 (A; V3 to V6; OF-MA, 1.35). For their part, the latter three had the worst SC-NMA values in their amplicon length categories (L, M, and M, respectively), and other alternatives within them, albeit targeting different regions, performed better *in silico*: KP_F048-OP_R030 (B, V3 to V7, 93.55%, 1.06) in the long category and OP_F053-KP_R020 (B, V1 to V3, 93.01%, 1.06) in the medium category. Consequently, the data derived from the primer combinations employed most in the literature could be improved upon, in some cases significantly, by using the alternative primer pairs presented in this study. Moreover, these results highlight that primer pairs targeting the same gene region do not distinguish equally between taxa with MAs.

Therefore, comparisons of data from studies assessing the same region may be biased, and abundance may be inaccurate but close. In the case of comparing amplicons from different regions, the results could be vastly different and may even lead to opposite biological conclusions. Consequently, the comparison of oral microbiome studies using the same primer pairs would be the most recommended methodological approach.

Our research proves that the detection of MAs with 100% sequence similarity between different taxa is not a one-off issue, as from 1.29% to 46.70% of the oral-prokaryotic species detected by the primer pairs have them. Indeed, this number may be an underestimate, given that we were only able to examine less than a third of the genomic sequences contained in the eHOMD (22), as the remaining ones were not fully sequenced. Despite this, relevant genera present in the oral environment were identified, including *Actinomyces*, *Fusobacterium*, *Lactobacillus*, *Methanosarcina*, *Staphylococcus*, and *Streptococcus* (16, 46), that had MAs in at least 10 primer pairs; 3.00%, 2.00%, 4.00%, 19.30%, 9.00%, and 15.00% of all different bacterial or archaeal species with MAs detected by all the primer pairs belonged, respectively, to such genera. Among them, there were health-associated species such as *S. mitis* (33) and *S. oralis* (34), disease-associated taxa such as *F. nucleatum* (30) and *S. mutans* (31), and those abundant in both states, such as *Methanosarcina vacuolata* (35), which, as shown above, had problems related to the presence of more than one 16S rRNA gene/genome. Other relevant species from distinct genera such as *Capnocytophaga ochracea*, *T. forsythia*, and *T. denticola* (30) also presented both intragenomic gene redundancy and MAs.

The main limitation of the present research is that only ~25% of the oral microorganism genomes listed on the eHOMD website were evaluated. This lack of complete genomes reduced the number of species evaluated to ~24% of those listed on eHOMD. Although the analysis could have been performed with a fasta file containing 16S rRNA gene sequences from oral microbes, we preferred to use complete genomes for the reasons stated previously (42), thus ensuring the high quality of the gene sequences reviewed. In our opinion, these results are only the tip of the iceberg, and the problematic issue of MAs is likely to affect more taxa as the number of genomes examined increases.

In conclusion, nearly all oral bacteria and about half of the oral archaea have more than one 16S rRNA gene in their respective genomes. Depending on the primer pair used, up to almost half of the species present MAs, affecting relevant genera present in the oral environment, such as *Actinomyces*, *Fusobacterium*, *Lactobacillus*, *Methanosarcina*, *Staphylococcus*, and *Streptococcus*. The performance of the primer pairs to detect non-MA species increases as the average length of the amplicons increases, none of these being the most widely used primer pairs in the oral microbiome literature. The best primer pairs were KP_F048-OP_R030 (for bacteria; region V3 to V7; primer pair position for *Escherichia coli* [GenBank version no. J01859.1], 342 to 1079), KP_F018-KP_R063 (for archaea; V3 to V9; undefined to 1506), and OP_F114-OP_R121 (for both bacteria and archaea; V3 to V9; 340 to 1405). In addition to the 16S rRNA gene redundancy, the considerable presence of MAs must be controlled to ensure an accurate interpretation of microbial diversity data. The SC-NMA is a more useful parameter than the conventional coverage percentage for selecting the best primer pairs. The choice of primer pair significantly affects diversity estimates and taxonomic classification, conditioning the comparability of oral microbiome studies using different primer pairs.

## MATERIALS AND METHODS

**Obtaining complete oral-bacterial and oral-archaeal genomes.** All the information available on the bacterial taxa present in the oral cavity was obtained from the eHOMD website (22). All genomes with the complete sequencing status indicated by eHOMD were chosen. A total of 528 complete genomes, consisting of one or more NCBI identifiers for each complete genome, were identified among 2,074 on the eHOMD website.

The complete genomes indicated in the eHOMD have one or more GenBank (25) identifiers, which were used to access the complete sequences stored in the NCBI database (26). In general, these complete genomes consisted of one or two identifiers corresponding to their chromosomal DNA; in many cases, however, the genomes had plasmid identifiers as well, which were also investigated.

An initial list of 177 different oral archaea and their corresponding GenBank (25) identifiers, obtained as part of a previous investigation (23), enabled us to access their complete sequences in the NCBI database (26). A total of 185 complete genomes, consisting of one or more NCBI identifiers for each complete genome, were identified.

Integrating the Entrez Programming Utilities (E-utilities) tool (47) in the Python (version 3.9.0) (http://www.python.org/) script allowed us to acquire the URLs needed to retrieve the information of interest from the various NCBI databases, including Taxonomy (48), RefSeq (49), and GenBank (25). The oral-bacterial and oral-archaeal genomes were then downloaded, and finally, the taxonomy of each of them was obtained.

**Detection and extraction of 16S rRNA genes.** There were several International Union of Pure and Applied Chemistry (IUPAC) ambiguous characters or nonspecific nucleotides distributed along some of the genomes. None of these characters or nucleotides represent a unique specification for the four nitrogenous bases of the DNA (A, adenine; G, guanine; C, cytosine; T, thymine) and allow for ambiguity among two, three, or four possible nucleic acid states (50). Consequently, we developed a Python

**TABLE 6** Selected primer pairs with high *in silico* coverage percentages targeting oral bacteria and/or archaea and those most used in the sequencing-based studies of the oral microbiome[a,b]

| ALC | F identifier | F Sequence 5-3 | F first post | F last post | R identifier | R sequence 5-3 | R first post | R last post | Length (bp) | Region |
|---|---|---|---|---|---|---|---|---|---|---|
| **Bacterial-specific primer pair** | | | | | | | | | | |
| S | KP_F048 | TACGGRAGGCAGCAG | 342 | 356 | OP_R043 | CCGCGRCTGCTGGCAC | 514 | 529 | 187 | V3–V4 |
| | OP_F098 | CCAGCAGCYGCGGTAAN | 517 | 533 | OP_R119 | GGACTACCRGGGTATCTAA | 787 | 805 | 288 | V4–V5 |
| | OP_F066 | GGMTTAGATACCC | 784 | 796 | KP_R040 | CCGTCAATTCMTTTGAGTTT | 906 | 925 | 141 | V5–V6 |
| | OP_F009 | GGATTAGATACCCBRGTAGTC | 784 | 804 | OP_R030 | TCACRRCACGAGCTGWCGAC | 1060 | 1079 | 295 | V5–V7 |
| | KP_F061 | ACTCAAAKGAATWGACGG | 908 | 925 | KP_R074 | GGGTYKCGCTCGTTR | 1099 | 1113 | 205 | V6–V7 |
| | OP_F101 | GAATTGRCGGGGRCC | 916 | 930 | OP_R030 | TCACRRCACGAGCTGWCGAC | 1060 | 1079 | 163 | V6–V7 |
| M | OP_F053 | GRGTTYGATYMTGGCTCAG | 9 | 27 | KP_R020 | CTGCTGCCTYCCGTA | 342 | 356 | 347 | V1–V3 |
| | KP_F048 | TACGGRAGGCAGCAG | 342 | 356 | KP_R031 | TACHVGGGTATCTAAKCC | 784 | 801 | 459 | V3–V5 |
| | KP_F048 | TACGGRAGGCAGCAG | 342 | 356 | OP_R073 | CRTACTHCHCAGGYG | 879 | 893 | 551 | V3–V6 |
| | KP_F051 | GTGCCAGCMGCNGCGG | 514 | 529 | KP_R041 | CGTCAATTCMTTTGAGTT | 907 | 924 | 410 | V4–V6 |
| | KP_F051 | GTGCCAGCMGCNGCGG | 514 | 529 | OP_R030 | TCACRRCACGAGCTGWCGAC | 1060 | 1079 | 565 | V4–V7 |
| | OP_F116 | YAACGAGCGCAACCC | 1099 | 1113 | KP_R060 | GACGGGCGGTGWGTRCA | 1390 | 1406 | 307 | V7–V9 |
| L | KP_F048 | TACGGRAGGCAGCAG | 342 | 356 | OP_R030 | TCACRRCACGAGCTGWCGAC | 1060 | 1079 | 737 | V3–V7 |
| | KP_F048 | TACGGRAGGCAGCAG | 342 | 356 | KP_R060 | GACGGGCGGTGWGTRCA | 1390 | 1406 | 1,064 | V3–V9 |
| | KP_F056 | AYTGGGYDTAAAGNG | 572 | 576 | KP_R077 | GACGGGCGGTGTGTACAA | 1389 | 1406 | 834 | V4–V9 |
| **Archaeal-specific primer pair** | | | | | | | | | | |
| S | KP_F018 | GYGCASCAGKCGMGAAW | U | U | KP_R002 | TTACCGCGGCKGCTG | 518 | 532 | | –V4 |
| | OP_F066 | GGMTTAGATACCC | 784 | 796 | KP_R013 | GGCCATGCACCWCCTCTC | U | U | | V5–V6 |
| M | KP_F018 | GYGCASCAGKCGMGAAW | U | U | KP_R032 | TACNVGGGTATCTAATCC | 784 | 801 | | V3–V5 |
| | KP_F018 | GYGCASCAGKCGMGAAW | U | U | OP_R073 | CRTACTHCHCAGGYG | 879 | 893 | | V3–V5 |
| | KP_F020 | CAGCMGCCGCGGTAA | 518 | 532 | KP_R013 | GGCCATGCACCWCCTCTC | U | U | | V3–V6 |
| | KP_F022 | AGGAATTGGCGGGGGAGCA | U | U | KP_R063 | TACCTTGTTACGACTT | 1491 | 1506 | | V5–V9 |
| L | OP_F114 | CCTAYGGGRBGCASCAG | 340 | 356 | KP_R013 | GGCCATGCACCWCCTCTC | U | U | | V3–V6 |
| | KP_F018 | GYGCASCAGKCGMGAAW | U | U | KP_R063 | TACCTTGTTACGACTT | 1491 | 1506 | | V3–V9 |
| | OP_F066 | GGMTTAGATACCC | 784 | 796 | OP_R016 | CGGTGTGTGCAAGGAG | U | U | | V5–V9 |
| **Bacterial and archaeal primer pair** | | | | | | | | | | |
| S | OP_F114 | CCTAYGGGRBGCASCAG | 340 | 356 | KP_R002 | TTACCGCGGCKGCTG | 518 | 532 | 192 | V3–V4 |
| | KP_F020 | CAGCMGCCGCGGTAA | 518 | 532 | KP_R032 | TACNVGGGTATCTAATCC | 784 | 801 | 283 | V4–V5 |
| | OP_F066 | GGMTTAGATACCC | 784 | 796 | OP_R073 | CRTACTHCHCAGGYG | 879 | 893 | 109 | V5–V6 |
| M | OP_F114 | CCTAYGGGRBGCASCAG | 340 | 356 | KP_R031 | TACHVGGGTATCTAAKCC | 784 | 801 | 461 | V3–V5 |
| | OP_F114 | CCTAYGGGRBGCASCAG | 340 | 356 | OP_R073 | CRTACTHCHCAGGYG | 879 | 893 | 553 | V3–V6 |
| | KP_F020 | CAGCMGCCGCGGTAA | 518 | 532 | OP_R073 | CRTACTHCHCAGGYG | 879 | 893 | 375 | V4–V6 |
| L | OP_F114 | CCTAYGGGRBGCASCAG | 340 | 356 | OP_R121 | ACGGGCGGTGWGTRC | 1391 | 1405 | 1,065 | V3–V9 |
| | KP_F020 | CAGCMGCCGCGGTAA | 518 | 532 | OP_R121 | ACGGGCGGTGWGTRC | 1391 | 1405 | 887 | V4–V9 |
| | OP_F066 | GGMTTAGATACCC | 784 | 796 | OP_R121 | ACGGGCGGTGWGTRC | 1391 | 1405 | 621 | V5–V9 |
| **Most used primer pair** | | | | | | | | | | |
| S | KP_F078 | GTGCCAGCMGCCGCGGTAA | 514 | 532 | OP_R010 | GGACTACHVGGGTWTCTAAT | 786 | 805 | 291 | V4–V5 |
| M | KP_F031 | AGAGTTTGATCCTGGCTCAG | 8 | 27 | KP_R021 | TTACCGCGGCTGCTGGCAC | 515 | 532 | 524 | V1–V4 |
| | KP_F047 | CCTACGGGNGGCWGCAG | 340 | 356 | KP_R035 | GACTACHVGGGTATCTAATCC | 784 | 804 | 464 | V3–V5 |
| | OP_F009 | GGATTAGATACCCBRGTAGTC | 784 | 868 | OP_R029 | ACGTCRTCCCCDCCTTCCTC | 1174 | 1193 | 409 | V5–V8 |
| L | KP_F014 | TCCAGGCCCTACGGG | U | U | KP_R011 | YCCGGCGTTGAMTCCAATT | U | U | | V3–V6 |
| | KP_F034 | AGAGTTTGATCMTGGCTCAG | 8 | 27 | KP_R065 | TACGGYTACCTTGTTACGACTT | 1491 | 1512 | 1,504 | V1–V9 |

[a]Primer pairs were selected based on the species coverage values (number of species detected/total species evaluated) in a previous investigation (23). They were individually evaluated through regular expressions against *Escherichia coli* (accession no. J01859) to define their positions. The U values represent a mismatch on the assessment and, therefore, the position cannot be confirmed with a guarantee. Gene regions were delimited as described by Baker et al. (54). In the column "gene region", the different regions covered by each primer pair are defined and delimited by a dash.

[b]ALC, amplicon length category; bp(s), base pair(s); F, forward; KP, Klindworth primer; L, long mean amplicon length category (>600 bp); M, medium mean amplicon length category (301 to 600 bp); OP, oral primer; post, position; R, reverse; S, short mean amplicon length category (100 to 300 bp); U, unaligned with *Escherichia coli*; V, hypervariable.

(http://www.python.org/) script to detect and then randomly replace them with one of the specific equivalent nucleotides (e.g., R was replaced by either A or G). This substitution was made to avoid alignment problems with the ambiguous nucleotides, and it was done on a random basis because we cannot know with certainty which of the options should have been assigned, as all the equivalents are correct. Other genomes were excluded because they had more than 10 consecutive ambiguous IUPAC nucleotides, mainly of "N" bases. After applying these criteria, we were left with 507 oral-bacterial complete genomes, of which 497 had one chromosome, nine had two chromosomes, and one had three chromosomes. Each individual chromosome was evaluated as a complete genome, so a final number of 518 oral-bacterial complete genomes were considered for analysis. Regarding the oral archaea, 177 complete genomes, of which 166 had one chromosome, 10 had two chromosomes, and one had five chromosomes, were left. Thus, a final number of 191 oral-archaeal complete genomes were considered for analysis.

Our Python (http://www.python.org/) script was completed with a free downloadable module known as search_16S_py (https://github.com/slyalina/search_16S_py), which is based on Edgar's algorithm (21). This algorithm looks for the 16S rRNA genes in the genomes, identifying sections with a high frequency of 13-mers in known 16S rRNA genes, and then, searches within each segment for conserved motifs close to the beginning and end of the gene. The obtention of a pair of motifs within the expected length range confirms the

presence of the gene and provides consistent and homologous endpoints (21). Applying this algorithm, the 16S rRNA gene sequences were detected and extracted from the complete downloaded genomes, while the variants were stored in a fasta file. All the 16S rRNA gene variants identified were designated taxonomically at the strain level or at the species level if no designated strain name existed. This left us with the following for inclusion in subsequent analyses: 518 oral-bacterial genomes, corresponding to 186 species, and 191 oral-archaeal genomes, corresponding to 135 species. Their taxonomy and NCBI identifiers are included in Table S10.

For each genome evaluated, we calculated its size, the sizes of the 16S rRNA genes detected, the total number of 16S rRNA genes, the number of different variants, and the number of 16S rRNA genes in each strand. The averages of the data obtained were subsequently determined using Python's NumPy (51) and pandas modules (52) for hierarchical levels above the strain level.

**Evaluation of a selection of primer pairs for the detection of 16S rRNA genes.** In the present study, primer pairs identified in a previous investigation as having the best *in silico* coverage values for detecting oral bacteria, oral archaea, or both (23) were selected for evaluation. Moreover, in that previous research, the authors found through a literature review that a total of 206 distinct primer pairs have been used to study the oral microbiome by massive sequencing (23). Those primers that were repeated more times (i.e., which appeared in more articles) were defined as those most frequently used in the oral microbiome literature and were also selected for evaluation here. This left us with 33 and 6 primer pairs, respectively, for this stage of the study, which were classified according to the average length of the amplicons into short (S; 100 to 300 bp), medium (M; 301 to 600 bp), and long (L; >600 bp) primer pairs (23) (Table 6).

The direct and reverse sequences of each primer pair were used in combination with Python's regex module (53) to obtain, *in silico*, the amplicons of the 16S rRNA genes identified in all of the chosen genomes. Both sequences were checked to ensure that they matched at some position in the genome evaluated and, if they did, all nucleotides from the first position of the direct primer to the last position of the reverse primer were selected to obtain the *in silico* amplicons. For each primer pair, we determined the mean size and number of the 16S rRNA gene amplicons, the number of gene variants, the number of genomes and species detected, and the SC-NMA. This coverage value was calculated as

$$SC-NMA\,(\%) = [(\text{Number of species detected} - \text{Number of different species with MAs})/$$

$$\text{Total number of species evaluated}] \times 100$$

The overestimation of abundance at the species level (the overestimation factor [OF]) was also calculated. This represented for each species the combination of the number of copies of the 16S rRNA gene amplicons and the number of MAs. To remove the overestimation derived from the intragenomic gene redundancy, the OF of each species was divided by the number of gene copies, resulting in the OF caused by the presence of MAs (OF-MA). Species with values equal to 1.00 did not have amplicons that matched other species for the corresponding primer pair, while those with estimates greater than 1.00 did. For each primer pair, both parameters were expressed cumulatively and as an average. The best primer pairs selected first were those with the highest SC-NMA value and, of these, those with the lowest OF-MA value. The worst primer pairs were those with the lowest SC-NMA and the highest OF-MA.

**Data availability.** The principal data generated or analyzed during this study are included in this published article. The Python script is publicly available at https://github.com/laravg/variant_analysis.

## SUPPLEMENTAL MATERIAL

Supplemental material is available online only.
**SUPPLEMENTAL FILE 1**, PDF file, 0.2 MB.
**SUPPLEMENTAL FILE 2**, XLSX file, 0.1 MB.
**SUPPLEMENTAL FILE 3**, XLSX file, 0.1 MB.
**SUPPLEMENTAL FILE 4**, XLSX file, 0.02 MB.
**SUPPLEMENTAL FILE 5**, XLSX file, 0.04 MB.
**SUPPLEMENTAL FILE 6**, XLSX file, 0.02 MB.
**SUPPLEMENTAL FILE 7**, XLSX file, 0.1 MB.
**SUPPLEMENTAL FILE 8**, XLSX file, 0.03 MB.
**SUPPLEMENTAL FILE 9**, XLSX file, 0.03 MB.
**SUPPLEMENTAL FILE 10**, XLSX file, 0.1 MB.

## ACKNOWLEDGMENTS

This study was funded by Instituto de Salud Carlos III (ISCIII) through project PI21/00588 and cofunded by the European Union, the Consellería de Cultura, Educación e Ordenación Universitaria de la Xunta de Galicia (accreditation 2019-2022 ED431G-2019/04, group with growth potential ED431B 2020-2022 GPC2020/27; A. Regueira-Iglesias support ED481A-2017/233), and the ERDF, which acknowledges the CiTIUS-Research Center in Intelligent Technologies of the Santiago de Compostela University as a Research Center of the Galician University System.

The funders had no role in the study design, data collection, analysis, decision to publish, or preparation of the manuscript.

C. Balsa-Castro and I. Tomás contributed to the conception and design of the study; A. Regueira-Iglesias and T. Blanco-Pintos searched the articles in PubMed and selected those of interest, extracting the relevant information; L. Vázquez-González, N. Vila-Blanco, and M. J. Carreira developed the bioinformatics procedures for obtaining the analysis; A. Regueira-Iglesias and C. Balsa-Castro made the graphs, tables, and additional files; A. Regueira-Iglesias and I. Tomás wrote the manuscript; M. J. Carreira carried out a critical review of the manuscript. All the authors approved the final version of the manuscript.

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
