## [Reviewer comments · Microbiology Spectrum]

Microbiology Spectrum

Impact of 16S rRNA gene redundancy and primer pair selection on the quantification and classification of oral microbiota in next-generation sequencing

Alba Regueira-Iglesias, Lara Vázquez-González, Carlos Balsa-Castro, Triana Blanco-Pintos, Nicolás Vila-Blanco, Maria José Carreira, and Inmaculada Tomás

Corresponding Author(s): Inmaculada Tomás, Oral Sciences Research Group, Special Needs Unit, Department of Surgery and Medical-Surgical Specialties, School of Medicine and Dentistry, Universidade de Santiago de Compostela, Health Research Institute Foundation of Santiago

Review Timeline:

Submission Date:	October 27, 2022
Editorial Decision:	November 19, 2022
Revision Received:	December 21, 2022
Accepted:	January 16, 2023

Editor: Jacqueline Abranches

Reviewer(s): The reviewers have opted to remain anonymous.

Transaction Report:

DOI: <https://doi.org/10.1128/spectrum.04398-22>

November 19, 2022

Prof. Inmaculada Tomás

Oral Sciences Research Group, Special Needs Unit, Department of Surgery and Medical-Surgical Specialties, School of Medicine and Dentistry, Universidade de Santiago de Compostela, Health Research Institute Foundation of Santiago (FIDIS) Santiago de Compostela Spain

Re: Spectrum04398-22 (Impact of 16S rRNA gene redundancy and primer pair selection on the quantification and classification of oral microbiota in next-generation sequencing)

Dear Prof. Inmaculada Tomás:

Link Not Available

Sincerely,

Jacqueline Abranches

Journals Department
Reviewer comments:

Reviewer #1 (Comments for the Author):

In this study, the authors examine the 16S rRNA genes in ~700 complete bacterial and archaeal genomes representing ~300 species. The authors examine how many copies of the 16S gene are in each genome, and examine sequence redundancy both within and across the genomes. They also examine the redundancy of amplicons from various primer sets. The study overall is logical and examines an important topic for oral microbiome research. There are a number of shortcomings in the current version of the document, mainly points that need clarification or should be explained in more detail to help with readability. Specific comments are as follows:

General: The specific variable regions covered by the various amplicons are not really mentioned in the text (just are included in Table 2), and I think certainly deserve some attention here, what V regions are the specific primer sets which are mentioned by name in the text amplifying?? I think this is important because many people are familiar with V1-V2 or V3-V4 nomenclature, for example, but not the primer names appearing in the text. Similarly, with full-length 16S rRNA sequencing now becoming more widely used with long-read sequencing advancements and ubiquity, it would be worth some speculation on how using full-length 16S analysis would compare to the best primer sets identified by the analyses here.

Line 47: "used against" please be more descriptive and specific, do you mean something like "aligned to predict amplicons"?

Line 43: It is somewhat unclear what is meant by "matching amplicons", and this is critical as it is clearly used throughout the text. I surmise it means redundant or identical amplicons, i.e. different species/strains with the same amplicon with a given primer set, so they can't be differentiated. Is this correct? It would be worthwhile to make sure that this is made crystal clear to the reader.

Lines 50-52: I'm not sure what the two percentages separated by a hyphen mean, what is the 2nd number? Please clarify...this format is used at multiple places in the text? ?

Line 65: You should probably be more specific and spell out "one intragenomic 16S rRNA gene" to avoid any confusion by the reader.

Line 119: Related to the above comment about matching amplicons, does "variants" here mean any difference in identity at all? (i.e. the genes must be 100% identical to be not be considered variants) or was there some other cutoff?

Line 119: So there were 186 discrete species among the 518 genomes? Spell this out more clearly right away.

Lines 141-142: It might be worth a sentence or two explaining how you selected which primer sets to examine. Similarly, it might be worth a sentence or two around Lines 116-120 explaining how you selected which genomes you were examining.

Lines 167-168: Although later in the text, you describe what you primers you are referring to when you say "commonly used", but please be specific about which commonly used primers you are referring to here and what their values were that did "stand out", so the reader isn't left wondering (better to define it here rather than later, or better yet, briefly in both places). "Stand out" is pretty general and should probably be made more specific.

Lines 214-219: This is the first time this tool is really mentioned in the manuscript, and should probably be introduced to some extent in the introduction and/or methods, particularly if the authors want to suggest in the Discussion that it is an innovative approach that itself raises the impact of the study.

Lines 225-228: A good point, but were the complete genomes available at the time of the previous studies to which the author is referring? And did the previous studies use tools that account for this? Or did the authors of the previous studies highlight that this was a limitation of their study? Probably better to mention any of these things and let the previous work/authors off the hook for the shortcoming at least a little.

Line 265: Should be "primer pairs with amplicons >600bp"?

Line 285: Again, please better define "used most" how was this determined, which primer sets did you consider to be "used most"?

Lines 308-310: Why were you only able to examine one third of the HOMD genomes? Ah, this is answered in Lines 320 and 321, please just move these facts up to Lines 308-310 so the reader isn't left wondering.

Lines 365-366: Please clarify what you mean here by IUPAC non-specific bases, just N's, or other wildcard nucleotides...Y, R, etc.? And also explain the choice to replace them randomly.

Line 364: Were the genomes examined not annotated? Why search for 16S rRNA genes here rather than extracting the 16S sequences from published annotations?

Lines 367-368: What was the cutoff for excluding genomes based on excess N's (or presumably other IUPAC non-specific bases)? More specificity is needed here.

Line 413: please mention where the Python script is publicly available.

Figure 1: It is not clear to me what is being shown here? Only species with matching amplicons are included? How did you determine what species are shown here? It is also not clear what the colors are denoting. By "# of primer pairs tested" do you mean the number of primer pairs tested for the given species, or the number of primer pairs tested for the given species that

gave matching amplicons?

Reviewer #2 (Comments for the Author):

I find the manuscript very interesting and well written. The different sections are well described, explained and presented. Overall, I think the work is relevant in the context of finding 16S redundancies across several oral biology associated species. I also believe that the primer selection part of the manuscript is lacking a more thorough development in connecting how these primers are linked to the hypervariable regions of the 16S gene and how would affect classification. Additionally, I think the authors could put more effort in the literature and add more content to the introduction, results and discussion. I found several publications that could be used to add weight to the manuscript and are completely missing from the references. Here I'm listing a few:

1. Uniting the classification of cultured and uncultured bacteria and archaea using 16S rRNA gene sequences.
2. A detailed analysis of 16S ribosomal RNA gene segments for the diagnosis of pathogenic bacteria.
3. The effect of 16S rRNA region choice on bacterial community metabarcoding results.

Specific comments on the text:

Line 117: It was said that the total number of oral bacteria examined was 518. Then in line 119 it was stated that a different number (186 bacterial species). The percentages latter presented were calculated based on the 186 total. It will make sense to the reader if the authors clarify the difference between these two counts (518 and 186) and clearly point the base count for the percentages calculations.

Staff Comments:

Preparing Revision Guidelines

Please return the manuscript within 60 days; if you cannot complete the modification within this time period, please contact me. If you do not wish to modify the manuscript and prefer to submit it to another journal, please notify me of your decision immediately so that the manuscript may be formally withdrawn from consideration by Microbiology Spectrum.

Dear Editor:

We submit the revised manuscript according to all suggestions made by both reviewers.

We would like to thank the reviewers for their comments, which have substantially improved the manuscript.

Below, we detail each of the reviewers' comments or questions and the corresponding response to each of them.

REVIEWER #1

Question 1. The specific variable regions covered by the various amplicons are not really mentioned in the text (just are included in Table 2), and I think certainly deserve some attention here, what V regions are the specific primer sets which are mentioned by name in the text amplifying??. I think this is important because many people are familiar with V1-V2 or V3-V4 nomenclature, for example, but not the primer names appearing in the text. Similarly, with full-length 16S rRNA sequencing now becoming more widely used with long-read sequencing advancements and ubiquity, it would be worth some speculation on how using full-length 16S analysis would compare to the best primer sets identified by the analyses here.

Response 1. Following your recommendations, we have incorporated next to each primer pair the variable regions that it amplifies, and this modification has been made throughout the manuscript. In addition, since this is how it is commonly represented in the literature, we have added the letter "V" for the variable region before the numbers indicating the region, and this modification has been made throughout the entire manuscript, including tables 2 and 3.

On the other hand, we have included some speculation on how using full-length 16S rRNA gene analysis would compare to the best primer sets identified by the analyses here. This information has been added to the Discussion section (lines 324 to 338).

Question 2. Line 47: "used against" please be more descriptive and specific, do you mean something like "aligned to predict amplicons"?

Response 2. By "used against" we meant "used to search for matches", and we have modified this point in the abstract. In the present work, we used regular expressions to search for matches of the direct and reverse sequences of the 39 primers in the whole genomes evaluated. If both sequences matched at some position within the genome, all nucleotides between the first position of the direct primer and the last of the reverse were selected to obtain *in silico* amplicons.

Due to the length limitations in the abstract, we have decided to expand the explanation of how *in silico* amplicons are obtained, being more descriptive and specific, in the manuscript body (Material and Methods section: lines 472 to 474).

Question 3. Line 43: It is somewhat unclear what is meant by "matching amplicons", and this is critical as it is clearly used throughout the text. I surmise it means redundant or identical amplicons, i.e. different species/strains with the same amplicon with a given primer set, so they can't be differentiated. Is this correct? It would be worthwhile to make sure that this is made crystal clear to the reader.

Response 3. Indeed, your assumption is somehow correct. We define "matching amplicons" as those amplicons with a 100% sequence similarity value that are obtained from different species using a given primer pair. However, by "redundant amplicons" we refer to the different 16S rRNA gene amplicons obtained from the same genome (i.e. the number of 16S rRNA gene amplicons per genome) using a given primer pair.

Clarifications have been added concerning these concepts the first time they are mentioned in the manuscript body (Introduction section: lines 94 to 97).

Question 4. Lines 50-52: I'm not sure what the two percentages separated by a hyphen mean, what is the 2nd number? Please clarify...this format is used at multiple places in the text?.

Response 4. The two percentages separated by a hyphen represent a range. In the sentence you point out: "Between 46.70%-1.29% of bacterial species and 38.89%-4.65% of archaeal detected by the primer pairs had MAs"; we want to indicate the maximum value of bacterial species detected by the tested primers showing matching amplicons (MAs) and the correspondent minimum value; as well as the maximum value of archaeal species detected by the tested primers showing MAs and the correspondent minimum value. To clarify the result we want to convey, the sentence has been amended in the abstract as follows: "Between 46.70% and 1.29% of bacterial species and among 38.89% and 4.65% of archaeal detected by the primers had MAs".

To clarify the wording and aid understanding of the article, we have also changed this format (a hyphen to separate a range) in the manuscript body (Results section: lines 174, 209, 213, 214, and 219; Discussion section: line 371). Still, we keep the hyphen to separate the start and end of gene regions (i.e., V3-V4 region), the start and end of amplicon length categories (i.e., 100-300 base pairs), numerical ranges within tables, as well as to separate words/abbreviations where necessary.

Question 5. Line 65: You should probably be more specific and spell out "one intragenomic 16S rRNA gene" to avoid any confusion by the reader.

Response 5. We incorporate your suggestion and indicate "one intragenomic 16S rRNA gene" in the "Importance" section of the article (line 65).

Question 6. Line 119: Related to the above comment about matching amplicons, does "variants" here mean any difference in identity at all? (i.e. the genes must be 100% identical to not be considered variants) or was there some other cutoff?

Response 6. In this research, variants of a gene (or of an amplicon) are defined as sequences that differ by at least one nucleotide from the reference sequence (the first sequence obtained). As you indicate, the genes (or amplicons) must be 100% identical in order not to be considered variants.

A clarification has been added with regard to this concept the first time it is mentioned in the manuscript body (Results section: lines 134 to 135).

Question 7. Line 119: So there were 186 discrete species among the 518 genomes? Spell this out more clearly right away.

Response 7. The total number of genomes evaluated is higher than the number of species found because the different strains of the same species have different genomes and, also, certain strains have more than one genomic identifier which could belong to chromosomal DNA or to plasmids. As can be seen in the supplementary file S10, there were species such as, for example, *Aggregatibacter actinomycetemcomitans* for which we found a total of eight different genomes belonging to eight different strains (S10.1); or *Methanosarcina barkeri* for which we found a total of five distinct genomes belonging to five distinct strains (S10.2). In addition, there were specific taxa such as *Ralstonia pickettii* DTP0602 (S10.1) and an undefined strain of *Sulfolobus acidocaldarius* (S10.2) for which we found more than one genomic identifier (specifically, a total of three chromosomes identifiers and five chromosomes identifiers, respectively).

Clarifications with respect to the discrepancy between the number of species and genomes have been added in the Results section (lines 126 to 130). Moreover,

modifications and clarifications regarding the total number of complete genomes downloaded and evaluated after applying the exclusion criteria have been added in the Material and Methods section

(lines 410 to 411, 418 to 419, and 437 to 442).

Question 8. Lines 141-142: It might be worth a sentence or two explaining how you selected which primer sets to examine. Similarly, it might be worth a sentence or two around Lines 116-120 explaining how you selected which genomes you were examining.

Response 8. Clarifications regarding how we selected the genomes and the primer pairs to examine have been added in the Results section (lines 124 to 126 and 157 to 160).

Question 9. Lines 167-168: Although later in the text, you describe what primers you are referring to when you say "commonly used", but please be specific about which commonly used primers you are referring to here and what their values were that did "stand out", so the reader isn't left wondering (better to define it here rather than later, or better yet, briefly in both places). "Stand out" is pretty general and should probably be made more specific.

Response 9. In this sentence, we specifically referred to the bacteria-specific primer pairs KP_F031-KP_R021, OP_F009-OP_R029, and KP_F034-KP_R065, that had the worst SC-NMA values (formerly described as "did not stand out") in their respective amplicon length categories (73.12%, 75.27%, and 82.26%, respectively). In the "archaea-specific" and "bacterial and archaeal" paragraphs of the Results section, the parameters obtained by the most commonly used primer pairs KP_F014-KP_R011 (archaea-specific) and KP_F078-OP_R010 (bacterial and archaeal) were already indicated, as well as it was defined that they showed the worst and second-worst SC-NMA values, respectively.

Furthermore, in the discussion section, we have made some modifications to clarify the ideas we want to convey: 1) to highlight that the majority of the “most used primers” were among those showing the worst SC-NMA values, and 2) to indicate that for all the most used primers in the oral microbiome literature we have found alternatives from the same amplicon length category and, sometimes, targeting the same region, with better *in silico* performance.

Clarifications regarding the results obtained by “the most commonly used primers” have been added in the Results section (lines 189 to 194) and discussion section (lines 344 to 360).

Question 10. Lines 214-219: This is the first time this tool is really mentioned in the manuscript, and should probably be introduced to some extent in the introduction and/or methods, particularly if the authors want to suggest in the Discussion that it is an innovative approach that itself raises the impact of the study.

Question 10. Indeed, following the rules of the structure of the journal (introduction, results, discussion, and material and methods), the first time we referred to the search_16S.py tool (Lyalina, 2019) was in the Discussion section. Although we had already included a detailed explanation of how the algorithm (Edgar, 2017) on which the tool is based works in the Material and Methods section (lines 444 to 448); following your recommendations, we have added a reference to the use of the search_16S.py tool in the introduction section (lines 117 to 119).

References:

Edgar R. SEARCH_16S: A new algorithm for identifying 16S ribosomal RNA genes in contigs and chromosomes. Preprint at bioRxiv 2017:124131. doi: 10.1101/124131.

Lyalina S. Search_16S_py_algorithm. 2019; Available at: https://github.com/slyalina/search_16S_py.

Question 11. Lines 225-228: A good point, but were the complete genomes available at the time of the previous studies to which the author is referring? And did the previous studies use tools that account for this? Or did the authors of the previous studies highlight that this was a limitation of their study? Probably better to mention any of these things and let the previous work/authors off the hook for the shortcoming at least a little.

Response 11. The articles we reference in lines 258 to 265 have publication dates of 2011, 2013, 2016, 2017, and 2018. According to the available eHOMD history, there were a total of 1,570 genomic DNA sequences in the database in 2019 (the earliest available date). Taking into account the current proportion of complete genomes in the database (25% of the total), there could have been around 392 at that point in time. Although it is true that in the more recent years there has been an exponential increase in the number of available genomes thanks to advances in sequencing technologies; it is also true that the intragenomic redundancy of the 16S rRNA gene is not a problem of recent years and its evaluation could have been carried out even in a smaller number of genomes.

Yet, as you point out, it was not an objective of the referenced research (and, to our knowledge, of any other study of the oral microbiome published to date) to account for or correct for 16S rRNA gene copy number variation between taxa, nor to highlight this issue as a study limitation. Thus, to let the previous work off the hook, we have added three sentences in lines 266 to 273 pointing this out, as well as the importance of future research using methods to correct for variation in gene copy number between taxa.

Question 12. Line 265: Should be "primer pairs with amplicons >600bp"?.

Response 12. That is correct, we are referring to primer pairs that obtain amplicons with mean lengths >600 base pairs, or primer pairs with long mean amplicon lengths.

Modifications have been made in this respect in the manuscript body (Results section: lines 179 to 183 and 198; Discussion section: lines 303 to 305 and 310 to 313).

Question 13. Line 285: Again, please better define "used most" how was this determined, which primer sets did you consider to be "used most"?

Response 13. In previous research of our investigation group (Regueira-Iglesias *et al.*, 2021), using a literature review, we found that a total of 206 different primer pairs sequences have been used to study the oral microbiota via massive sequencing. Those that were repeated more times (i.e., which appeared in more articles) were: KP_F078-OP_R010 (33 articles), KP_F047-KP_R035 (21), KP_F014-KP_R011 (8), KP_F034-KP_R065 (8), KP_F031-KP_R021 (7), and OP_F009-OP_R029 (7); and they were identified as “the most commonly used primer pairs in the oral microbiome studies”. On the other hand, four, three, four, 10, and 21 distinct primer pairs were repeated six, five, four, three, and two times; and, lastly, 158 were found only once.

Clarifications regarding what is meant by “most commonly used primer pairs” and where/how they were obtained have been added in the Results section (lines 158 to 160) and Material and Methods (lines 463 to 466) section.

Reference:

Regueira-Iglesias A, Vázquez-González L, Balsa-Castro C, Vila-Blanco N, Blanco-Pintos T, Tamames J, Carreira MJ, Tomás I. *In silico* evaluation and selection of the best 16S rRNA gene primers for use in next-generation sequencing to detect oral bacteria and archaea. Accepted for publication in Microbiome. Preprint at Research Square. 2021. doi: 10.21203/rs.3.rs-516961/v1.

Question 14. Lines 308-310: Why were you only able to examine one third of the HOMD genomes? Ah, this is answered in Lines 320 and 321, please just move these facts up to Lines 308-310 so the reader isn't left wondering.

Response 14. We incorporate your suggestion and move this explanation to the end of the phrase “Indeed, this number may be an underestimate, given that we were only able to examine less than a third of the genomes contained in the eHOMD” (Discussion section: lines 373 to 374).

Question 15. Lines 365-366: Please clarify what you mean here by IUPAC non-specific bases, just N's, or other wildcard nucleotides...Y, R, etc.? And also explain the choice to replace them randomly.

Response 15. By IUPAC non-specific nucleotides we refer to all the ambiguous characters that do not represent a unique specification for the four nitrogenous bases of the DNA (A= adenine, G= guanine, C= cytosine, T= thymine): R, Y, S, W, K, M, B, D, H, V, N, and “.” or “-“; and allow for ambiguity among two, three, or four possible nucleic acid states (Johnson, 2010). Thus, for example, R is used for unspecified purine bases (A or G) and Y is used for unspecified pyrimidine bases (C or T). If these ambiguous characters were not replaced, we would have alignment problems.

On the other hand, we chose to randomly replace these ambiguous nucleotides with one of the specific equivalents (e.g. R was replaced either by A or G) because we cannot know with certainty which of the options should have been assigned as all the equivalents were correct and, as said before, they had to be substituted to avoid alignment problems.

Clarifications regarding what we meant by non-specific bases, together with a corresponding citation, and the reasons for random substitution have been added to the Material and Methods section (lines 427 to 435).

Reference:

Johnson AD. An extended IUPAC nomenclature code for polymorphic nucleic acids. *Bioinformatics*. 2010 May;26(10):1386-1389.

Question 16. Line 364: Were the genomes examined not annotated? Why search for 16S rRNA genes here rather than extracting the 16S sequences from published annotations?

Response 16. All genomes evaluated were stored in the NCBI databases (GenBank, Taxonomy, and RefSeq) and these are annotated. However, we did not use any 16S rRNA gene sequence annotations, we only downloaded the fully assembled complete genome sequences.

Although the analysis could have been performed on annotations of the 16S rRNA gene sequences from oral microbes (from this or other databases); we preferred to use complete genomes, thereby ensuring the high quality of the sequences reviewed due to the following reasons:

- 1) Edgar (Edgar, 2018) estimated that the taxonomy annotation error rate of the Ribosomal Database Project (RDP) database is ~10%; on the other hand, he found 249,490 identical sequences with conflicting annotations in SILVA v128 (Quast *et al.*, 2013) and Greengenes v13.5 (DeSantis *et al.*, 2006) at ranks up to phylum (7,804 conflicts), indicating that the annotation error rate in these databases is ~17%.
- 2) We have verified in previous research that a very high percentage of 16S rRNA gene annotations present a loss of information of up to 60-70 nucleotides in regions 1 and 9 of the sequences, which invalidates their use (Regueira-Iglesias *et al.*, 2021)
- 3) Most of the complete genomes evaluated in our study are isolates that were sequenced with Sanger technology or with second-generation technology (shorter sequences than Sanger). In both cases, contig scaffolding algorithms were used to construct the complete genomes from the sequences with a minimum coverage of 8x for Sanger sequences and 30x in the case of second-generation technologies (Schatz *et al.*, 2010). In these types of assemblies, positions within the genome that did not have high

coverage included non-specific nucleotides. In the present investigation, we discarded genomes that included more than 10 consecutive unspecific positions.

4) Lastly, many genomes were downloaded from the NCBI RefSeq database (O'Leary *et al.*, 2016), where the annotations of the complete genomes were manually curated or re-annotated concerning the information provided by the original author, including their taxonomic hierarchy.

All these reasons led us to prefer the use of complete genomes and, from these, to detect and extract 16S rRNA genes using the above-mentioned search_16S_py tool (Lyalina, 2019) based on Edgar's algorithm (Edgar, 2017) which has an estimated sensitivity >99% for identifying known 16S rRNA genes. Thus, the results reported here highlight only part of a much more extensive problem.

According to these explanations, the sentences in lines 388 and 389 (Discussion section) have been completed.

References:

DeSantis TZ, Hugenholtz P, Larsen N, Rojas M, Brodie EL, Keller K, Huber T, Dalevi D, Hu P, Andersen GL. Greengenes, a chimera-checked 16S rRNA gene database and workbench compatible with ARB. *Appl Environ Microbiol.* 2006 Jul;72(7):5069-5072.

Edgar R. SEARCH_16S: A new algorithm for identifying 16S ribosomal RNA genes in contigs and chromosomes. Preprint at bioRxiv. 2017:124131. doi: 10.1101/124131.

Edgar R. Taxonomy annotation and guide tree errors in 16S rRNA databases. *PeerJ.* 2018 Jun;6:e5030. doi: 10.7717/peerj.5030.

Lyalina S. Search 16S py algorithm. 2019; Available at: https://github.com/slyalina/search_16S_py.

O'Leary NA, Wright MW, Brister JR, Ciuffo S, Haddad D, McVeigh R, Rajput B, Robbertse B, Smith-White B, Ako-Adjei D, Astashyn A, Badretdin A, Bao Y, Blinkova

O, Brover V, Chetvernin V, Choi J, Cox E, Ermolaeva O, Farrell CM, Goldfarb T, Gupta T, Haft D, Hatcher E, Hlavina W, Joardar VS, Kodali VK, Li W, Maglott D, Masterson P, McGarvey KM, Murphy MR, O'Neill K, Pujar S, Rangwala SH, Rausch D, Riddick LD, Schoch C, Shkeda A, Storz SS, Sun H, Thibaud-Nissen F, Tolstoy I, Tully RE, Vatsan AR, Wallin C, Webb D, Wu W, Landrum MJ, Kimchi A, Tatusova T, DiCuccio M, Kitts P, Murphy TD, Pruitt KD. Reference sequence (RefSeq) database at NCBI: current status, taxonomic expansion, and functional annotation. *Nucleic Acids Res.* 2016 Jan;44(D1):D733-D745. doi: 10.1093/nar/gkv1189.

Quast C, Pruesse E, Yilmaz P, Gerken J, Schweer T, Yarza P, Peplies J, Glöckner FO. The SILVA ribosomal RNA gene database project: improved data processing and web-based tools. *Nucleic Acids Res.* 2013 Jan;41(Database issue):D590-D596.

Regueira-Iglesias A, Vázquez-González L, Balsa-Castro C, Vila-Blanco N, Blanco-Pintos T, Tamames J, Carreira MJ, Tomás I. *In silico* evaluation and selection of the best 16S rRNA gene primers for use in next-generation sequencing to detect oral bacteria and archaea. Accepted for publication in *Microbiome*. Preprint at Research Square. 2021. doi: 10.21203/rs.3.rs-516961/v1.

Schatz MC, Delcher AL, Salzberg SL. Assembly of large genomes using second-generation sequencing. *Genome Res.* 2010 Sep;20(9):1165-1173.

Question 17. Lines 367-368: What was the cutoff for excluding genomes based on excess N's (or presumably other IUPAC non-specific bases)? More specificity is needed here.

Response 17. Indeed, some genomes were excluded for having more than 10 consecutive ambiguous IUPAC nucleotides; not only N's but R, Y, S, W, K, M, B, D, H, V, N, and "." or "-".

Specifications regarding these genome exclusion criteria have been added in the Material and Methods section (line 436).

Question 18. Line 413: please mention where the Python script is publicly available.

Response 18. Information regarding the public repository where our Python script is located has been added to the Data availability section (line 495).

Question 19. Figure 1: It is not clear to me what is being shown here? Only species with matching amplicons are included? How did you determine what species are shown here? It is also not clear what the colors are denoting. By "# of primer pairs tested" do you mean the number of primer pairs tested for the given species, or the number of primer pairs tested for the given species that gave matching amplicons?

Response 19. Figure 1 represents all oral species (23 bacterial -A- and 17 archaeal -B-) for which at least 10 of the tested primer pairs found that these species had MAs. Therefore, yes, we are only representing species with MAs; and we selected all the species that had MAs using at least 10 of the tested primer pairs.

Regarding the colour appreciation, in both panels of the figure, each number of primer pairs was given a colour, so 10 primer pairs correspond to electric green, 11 to violet, 12 to grey, 13 to black, and so on. Different colours were used for each number to facilitate visual differentiation, and these were chosen purely for aesthetic reasons.

Concerning your last question, your second assumption is correct and by "No. of primer pairs tested" we mean the number of primer pairs tested that found that a given species had MAs. Thus, for example, 10 of the tested primer pairs found that *Fusobacterium nucleatum* had MAs, and a total of 25 obtained that *Streptococcus oralis* and *Streptococcus mitis* had MAs.

REVIEWER 2

Question 1. I think the authors could put more effort in the literature and add more content to the introduction, results and discussion. I found several publications that could be used to add weight to the manuscript and are completely missing from the references. Here I'm listing a few: 1. Uniting the classification of cultured and uncultured bacteria and archaea using 16S rRNA gene sequences; 2. A detailed analysis of 16S ribosomal RNA gene segments for the diagnosis of pathogenic bacteria; 3. The effect of 16S rRNA region choice on bacterial community metabarcoding results.

Response 1. The investigations recommended by you have been studied and references to them have been incorporated where appropriate into the Introduction section (lines 90 to 93) and Discussion section (lines 289 to 290, line 326, and lines 329 to 330) of the manuscript. In addition, throughout the entire manuscript, we have developed in detail concepts included in the previous version as well as added new ones and their corresponding references (Introduction section: lines 90 to 97 and 115 to 119; Results section: lines 124 to 130, 134 to 135, 157 to 160, and 189 to 194; Discussion section: lines 243 to 245, 266 to 273, 289 to 290, 324 to 338, 344 to 360, and 370 to 374; Material and Methods section: lines 410 to 411, 418 to 419, 427 to 442, 461 to 466, and 472 to 474).

Question 2. Line 117: It was said that the total number of oral bacteria examined was 518. Then in line 119 it was stated that a different number (186 bacterial species). The percentages latter presented were calculated based on the 186 total. It will make sense to the reader if the authors clarify the difference between these two counts (518 and 186) and clearly point the base count for the percentages calculations.

Response 2. Following your recommendations, clarifications with respect to the discrepancy between the number of species and genomes have been added in the Results section (lines 126 to 130). As we explain there, the total number of genomes

evaluated is higher than the number of species found because the different strains of the same species have different genomes and, also, certain strains have more than one genome identifier which could belong to chromosomal DNA or plasmids. As can be seen in the supplementary file S10, there were species such as, for example, *Aggregatibacter actinomycetemcomitans* for which we found a total of eight different genomes belonging to eight different strains (S10.1); or *Methanosarcina barkeri* for which we found a total of five distinct genomes belonging to five distinct strains (S10.2). In addition, there were specific taxa such as *Ralstonia pickettii* DTP0602 (S10.1) and an undefined strain of *Sulfolobus acidocaldarius* (S10.2) for which we found more than one genome identifier (specifically, a total of three and five, respectively).

The percentages shown in these two paragraphs of the Results section (lines 136 to 151) were calculated based on the 186 total number of bacterial species and the 135 total number of archaeal species because we are referring to the data (number of genes, variants...) obtained at the species level. Again, following your recommendations, clarifications with respect to the base count for the percentages calculations have been added in the Results section (lines 136 to 151, 171 to 175, and 213 to 214).

January 16, 2023

Prof. Inmaculada Tomás

Oral Sciences Research Group, Special Needs Unit, Department of Surgery and Medical-Surgical Specialties, School of Medicine and Dentistry, Universidade de Santiago de Compostela, Health Research Institute Foundation of Santiago
Surgery and Medical-Surgical Specialties
Santiago de Compostela
Spain

Re: Spectrum04398-22R1 (Impact of 16S rRNA gene redundancy and primer pair selection on the quantification and classification of oral microbiota in next-generation sequencing)

Dear Prof. Inmaculada Tomás:

Your manuscript has been accepted, and I am forwarding it to the ASM Journals Department for publication. You will be notified when your proofs are ready to be viewed.

Sincerely,

Jacqueline Abranches
Editor, Microbiology Spectrum
